# Characterisation of the immune repertoire of a humanised transgenic mouse through immunophenotyping and high-throughput sequencing

Eve Richardson[1,2†], Špela Binter[1†], Miha Kosmac[1], Marie Ghraichy[3,4], Valentin von Niederhäusern[3,4], Aleksandr Kovaltsuk[2], Jacob D Galson[5], Johannes Trück[3,4], Dominic F Kelly[6], Charlotte M Deane[2*], Paul Kellam[1,7], Simon J Watson[1*]

[1]Kymab, a Sanofi Company, Babraham Research Campus, Cambridge, United Kingdom; [2]Department of Statistics, University of Oxford, Oxford, United Kingdom; [3]Division of Immunology, University Children's Hospital, University of Zurich, Zurich, Switzerland; [4]Children's Research Center, University of Zurich, Zurich, Switzerland; [5]Alchemab Therapeutics Ltd, Kings Cross, London, United Kingdom; [6]Department of Paediatrics, University of Oxford, Oxford, United Kingdom; [7]Department of Infectious Disease, Faculty of Medicine, Imperial College London, London, United Kingdom

**\*For correspondence:**
deane@stats.ox.ac.uk (CMD);
Simon.Watson@rqbiotechnology.com (SJW)

†These authors contributed equally to this work

## Abstract

Immunoglobulin loci-transgenic animals are widely used in antibody discovery and increasingly in vaccine response modelling. In this study, we phenotypically characterised B-cell populations from the Intelliselect Transgenic mouse (Kymouse) demonstrating full B-cell development competence. Comparison of the naïve B-cell receptor (BCR) repertoires of Kymice BCRs, naïve human, and murine BCR repertoires revealed key differences in germline gene usage and junctional diversification. These differences result in Kymice having CDRH3 length and diversity intermediate between mice and humans. To compare the structural space explored by CDRH3s in each species' repertoire, we used computational structure prediction to show that Kymouse naïve BCR repertoires are more human-like than mouse-like in their predicted distribution of CDRH3 shape. Our combined sequence and structural analysis indicates that the naïve Kymouse BCR repertoire is diverse with key similarities to human repertoires, while immunophenotyping confirms that selected naïve B cells are able to go through complete development.

## Editor's evaluation

This is an important study that defines the limits of using human Ig sequences in genetically modified mice to explore human immune responses. The study was carefully designed, and the results should be of interest to a wide readership and will be important for grounding future studies/uses of the Ky mouse.

## Introduction

Twenty-five years of progress in genetic engineering from the first immunoglobulin (Ig) transgenic mouse (*Brüggemann et al., 1989*) culminated in 2014 in the integration of a complete human Ig

haplotype in mice for the first time (*Lee et al., 2014*). Humanised Ig loci-transgenic animal models have proven extremely useful in therapeutic antibody discovery; 20 of the 127 therapeutic antibodies licensed in the US or EU as of April 2022 were derived from transgenic mouse platforms (data from Thera-SAbDab; *Raybould et al., 2020*). Transgenic platforms have also found a new application in vaccine response modelling (*Sok et al., 2016*; *Pantophlet et al., 2017*; *Walls et al., 2020*). As humanised animal models become the source of a growing number of therapeutics and play an increasingly important role in the evaluation of novel vaccine candidates, it is crucial to understand the degree to which their B-cell repertoires can be considered representative of humans.

Contemporary Ig transgenic animal models vary according to the number of genes and localisation of the inserted human Ig loci (*Green, 2014*; *Brüggemann et al., 2015*). In Kymab's Intelliselect Transgenic mouse (Kymouse), a complete set of human variable (V), diversity (D), and junction (J) genes of the IGH locus as well as the V and J genes of the Ig $\lambda$ and Igκ loci were inserted at the sites of the endogenous mouse loci. The mouse constant regions were retained, preserving downstream interactions with endogenous intracellular signalling components and cell membrane Fc receptors, resulting in functional, fully active chimeric antibodies. Kymice exhibit normal B-cell production and maturation and the resulting B-cell receptors (BCRs) are diverse, with human-like CDRH3 lengths and evidence of somatic hypermutation (*Lee et al., 2014*). However, the baseline phenotypic diversity in B cells and BCRs in the Kymouse has not been fully described.

B cells are an integral part of the humoral immune response due to their ability to produce antibodies against diverse antigens, providing protection against infection. B cells originate from hematopoietic stem cells in the bone marrow, where they undergo several phases of antigen-independent development leading to the generation of immature B cells. B cells are routinely classified based on their maturation status, antibody isotype, and effector function. Ig gene rearrangement during these early stages of B-cell development results in the expression of a mature BCR that is capable of binding to antigen. This is followed by positive and negative selection processes, to eliminate non-functional and self-reactive immature B cells. Surviving B cells complete antigen-independent maturation in the spleen, producing immunocompetent naïve mature B cells that subsequently develop into either follicular or marginal zone B cells. In response to vaccination or invading microbes, antigen-specific B cells within secondary lymphoid organs differentiate into antibody-producing cells, early memory cells, or rapidly proliferate and form structures known as germinal centres (GCs) (*Allen et al., 2007*). GCs are inducible lymphoid microenvironments that support the generation of affinity-matured, isotype-switched memory B cells and antibody-secreting plasma cells. Long-lived plasma cells secrete high-affinity antibodies, and memory cells can readily elicit an efficient antibody immune response upon re-exposure to the immune stimuli (*Corcoran and Tarlinton, 2016*; *Weisel and Shlomchik, 2017*). Iterative cycles of B-cell hypermutation and selection within the GC leads to an accumulation of affinity-enhancing mutations and ultimately to the progressive increase of serum antibody affinity, a process known as antibody affinity maturation (*Jacob et al., 1991*). Antibody-secreting plasma cells play critical roles in protective immunity on the one hand and antibody-mediated autoimmune disease on the other. During immune responses a small fraction of newly generated plasma cells enter either the bone marrow or the lamina propria of the small intestine where they populate specialised survival niches and become long-lived plasma cells (*Lemke et al., 2016*) thus maintaining antibody titres for extended periods.

The variable domain of a BCR is composed of a heavy chain and a light chain. Each of the chains in the antibody has three hypervariable regions known as the complementarity-determining regions (CDRs), which make most contacts with the antigen. The heavy chain locus consists of variable (V), diversity (D), and joining (J) gene segments, which recombine to form the variable domain of the heavy chain ($V_H$). These genes are referred to as the IGHV, IGHD, and IGHJ genes, respectively. The first two CDRs of the heavy chain, CDRH1 and CDRH2, are encoded by the IGHV gene alone, while the third and most variable CDR, CDRH3, spans the IGHV, IGHD, and IGHJ gene junctions. The insertion of random and palindromic nucleotides at the VD and DJ junctions further contributes to the diversity of the CDRH3, ensuring binding diversity to different antigens and epitopes (*Xu and Davis, 2000*). Each of the light chain loci, kappa and lambda, consist of V and J gene segments but no D gene segments, and both the germline and the recombined light chain variable region ($V_L$) are less diverse than their heavy chain counterparts (*Collins and Watson, 2018*). These genes are referred to as the IGKV and IGKJ or IGLV and IGLJ genes for the kappa and lambda

chains, respectively: we use IGKLV or IGKLJ to refer to the V or J genes of either light chain locus collectively.

Due to the greater diversity of the heavy chain, most next-generation sequencing (NGS) of BCR repertoires (BCR-seq) has focused on the heavy chain; lower throughput methods exist for identifying the light chain pairing (*Curtis and Lee, 2020*). The resulting BCR sequences can be aligned to reference germline gene databases to infer most likely germline gene origins and insertions or deletions at the V(D) or (D)J junctions (*Ye et al., 2013*). Alignment of BCRs to common germline genes also allows inference about clonal structure, as sequences sharing common germline gene assignments as well as homology in the CDRH3 loop may be inferred to have arisen from a common progenitor B cell (*Greiff et al., 2015*; *Yaari and Kleinstein, 2015*). The amino acid sequences of these heavy chains can also be functionally examined through annotation with structural tools (*Kovaltsuk et al., 2017*; *Marks and Deane, 2020*). Changes in the pattern of CDRH3 shapes in BCR repertoires have been observed along the B-cell differentiation axis in both humans and mice (*Kovaltsuk et al., 2020*) but the extent to which the CDRH3 shape differs between humans and mice has not been explored.

Here, we have characterised the frequency of GC B cells, memory B cells, and long-lived plasma cells from spleens, lymph nodes, and bone marrows of antigen-inexperienced Kymice (*Lee et al., 2014*). The frequencies of these B-cell subsets as well as the breadth and nature of their BCR repertoires constitute the first step in our understanding of how the immune system of this model organism responds to different antigens, vaccines, and pathogens that are both of scientific and of therapeutic interest. Examining the nature of the naïve BCR repertoire in Kymice through both single-cell and bulk sequencing and structural analysis shows that the Kymouse naïve BCR repertoires are more human-like in their distribution of CDRH3 shapes.

## Results

### Antigen naïve Kymice exhibit similar B-cell subpopulation frequencies

We characterised the B-cell subpopulations within spleen and lymph node samples of 12 antigen-inexperienced Kymice using an 11-colour flow cytometry panel that incorporated a range of B-cell lineage markers to identify both murine memory and GC B-cell populations. A canonical gating scheme organises B cells by their maturation status – from transitional B cells through naïve, non-switched and ultimately class-switched memory B cells. To look at the heterogeneity of the B-cell subpopulations in more detail, we incorporated unbiased Leiden clustering on the multi-parameter fluorescently activated cell sorting (FACS) data. Sorted cells separated into two large clusters, B cells and non-B cells (*Figure 1A*). As expected, within the B-cell population immature isotypes (IgD and IgM) were enriched in naïve cells, whereas markers CD95 and GL7 were enriched in GC cells (*Figure 1C*). The murine memory B-cell population has been described to comprise five subpopulations defined by the progressive transition from naive-like to more memory-like cells and the surface markers CD80, CD73, and PD-L2 have previously been reported to enable their distinction (*Tomayko et al., 2010*). Using a low dimensional UMAP representation, we observed distinct staining patterns of these markers in the memory B-cell compartment and were able to distinguish between 12 major B-cell populations, including transitional, naïve, and activated as well as six distinct memory subsets, defined as (1) PD-L2$^{hi}$, (2) CD73$^{hi}$ CD80$^{hi}$ PD-L2$^{low}$, (3) CD80$^{low}$, (4) PD-L2$^{hi}$ CD80$^{hi}$ CD73$^{low}$, (5) CD80$^{hi}$, and (6) CD73$^{hi}$ (*Figure 1A*). GC B cells formed a small and well-separated cluster whose small frequency was not surprising given that these were antigen naïve animals. Based on the three memory markers (CD80, CD73, and PD-L2) the relative frequency of the total memory B cells was 6.60% ± 2.51%, and the frequency of CD95 and GL7 positive GC cells was 0.18% ± 0.26%. The median expression profile of each subset is shown as a heatmap (*Figure 1B*). The antigen naïve B-cell populations in un-immunised and non-infected Kymice are therefore normal and consistent between different Kymice (*Figure 1B*, right panel).

### B cells in the bone marrow are class switched with variable levels of surface BCR expression

To understand the B-cell profile beyond spleen and lymph nodes, we also profiled the bone marrow cells of mice. We characterised bone marrow samples using a nine-colour flow cytometry panel that incorporated a range of B-cell lineage markers. The staining panel was designed to identify plasma

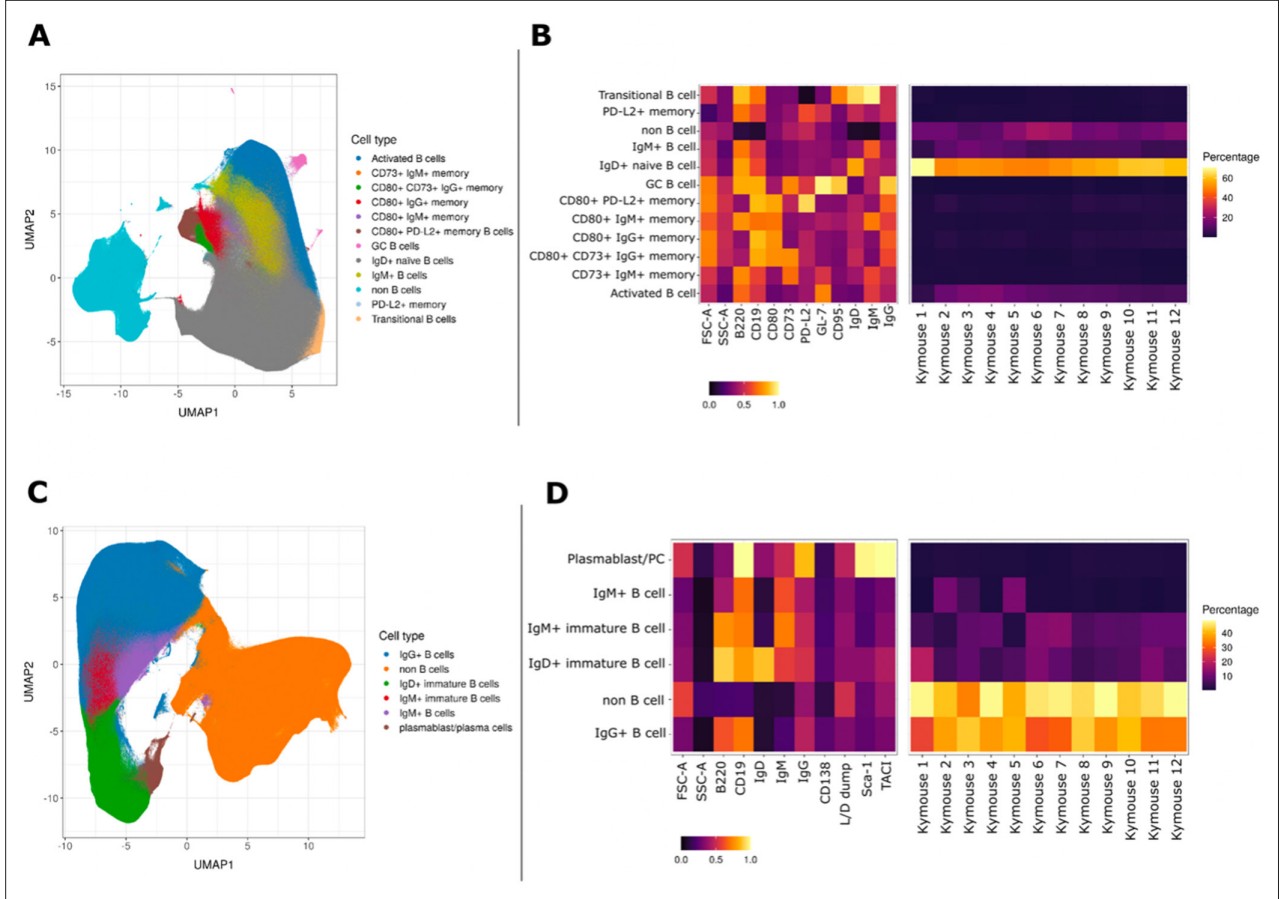

**Figure 1.** UMAP projections of sorted cell populations from the spleen, lymph nodes and bone marrow of Kymice. UMAP projections of sorted cell populations identified using unsupervised clustering from spleen and lymph nodes (**A**) or from bone marrow samples (**C**) can be used to visualise marker expression on the combined cells that were used for sorting and to characterise their phenotypes. UMAP projections show a clear separation between B cells and non-B cells for both sample types. The projections are coloured by the 12 resolved cell types in the spleen and lymph nodes (**A**) and the six resolved in the bone marrow samples (**C**). Normalised and scaled marker expression and frequencies were used to visualise mouse-to-mouse variation for each of the resolved cell types in the spleen and lymph nodes (**B**) or the bone marrow samples (**D**). The expression profiles are homogeneous across mice. In spleens and lymph nodes non-class-switched IgD+ naive B cells were the predominant cell population, followed by non-B cells and IgM+ naive B cells, reflective of a tissue that has not been exposed to antigen. In bone marrow samples on the other hand the most numerous populations were non-B cells, followed closely by class-switched IgG+ B cells, a result that reflects a tissue niche that supports survival of long-lived antibody producing cells.

cells as well as class-switched B-cell subsets in antigen-inexperienced mice. As expected, we saw that the cells separated again into two large clusters, B cells and non-B cells (*Figure 1C*). The expression profiles of the subsets were again plotted as heatmaps showing the median expression profiles of each subset (*Figure 1D*). Within the B-cell cluster, we identified several discrete subsets marked by the expression of different BCR isotypes. B cells were clustered into five distinct subpopulations, including immature IgD+ and IgM+ cells, mature IgM+ and IgG+ B cells, and plasma cells. The markers TACI and Sca-1 were enriched in plasma cells as expected, whereas CD138, a common plasma cells marker did not show a high level of separation between the different cell types. Unsurprisingly, we saw the biggest separation between B cells and non-B cells, and a continuum of B-cell subtypes from IgD, through IgM, to IgG-expressing B cells as well as a discrete cluster identified as plasma cells. The frequencies of the plasma cells were low (0.90% ± 0.28%) in comparison to other B-cell subtypes, perhaps not surprising given that these were antigen naïve animals.

## The Kymouse naïve antibody sequence repertoire is more human-like than murine-like

Using high-throughput paired sequencing we recovered 3175 full-length paired IgM $V_H$ and $V_L$ sequences and a further 451,655 full-length unpaired IgM $V_H$ sequences from naïve B cells extracted from the spleens and lymph nodes of 22 Kymice.

In order to evaluate the humanness of the Kymouse naïve B-cell sequence repertoire, we performed two comparisons: a comparison of the lower-depth single-cell paired sequencing data with a published, high-depth paired human naïve dataset, and a more extensive comparison of the bulk $V_H$ sequences to equivalent datasets of 338,677 $V_H$ sequences from human naive B cells, and 268,285 $V_H$ sequences from C57BL/6 mice.

## Paired $V_H$ and $V_L$ sequencing suggests that Kymice produce primary repertoires with differing germline gene usages than humans

One of the most pronounced differences in heavy/light chain pairing between wild-type mice and humans that has been described is the usage ratio of the Igκ and Ig $\lambda$ chains in the BCRs of circulating B cells. Humans have an Igκ/Ig $\lambda$ ratio of approximately 60:40 in serum and in mature B cells: the Igκ/Ig $\lambda$ ratio in the human naïve single-cell dataset was 62:38 (IQR: 64:36 59:41). Mice have an Igκ/Ig $\lambda$ ratio of 95:5 in serum and 90:10 on B cells (*McGuire and Vitetta, 1981*). We used the 3175 paired $V_H$ and $V_L$ sequences to calculate the Igκ/Ig $\lambda$ ratio in Kymice and found a ratio of 51:49 (IQR: 55:45, 47:53), which is considerably closer to the human ratio, as both reported in the literature and measured in the human paired dataset, than the mouse ratio.

We next analysed the heavy and light chain gene usage in the paired data and compared the observed frequencies with those observed in the large paired human dataset (*Figure 2*). There are differences in usage of all gene segments on both chains: IGHV genes of subgroup IGHV3 and IGHV6 are expressed at a higher rate in Kymouse vs human repertoires (an average of 66.4% vs 44.8% and 4.0% vs 0.7% of Kymouse and human repertoires, respectively), with a decreased usage of IGHV1 and IGHV4 genes relative to human repertoires (an average of 10.5% vs 19.4% and 15.8% vs 26.1% of Kymouse and human repertoires, respectively) (*Figure 2A*). The IGHD gene subgroup usage was likewise different with increased usage of IGHD1, IGHD6, and IGHD7 in the Kymouse, and a significantly reduced usage of IGHD2, IGHD4, and IGHD5 (*Figure 2B*). The Kymice use IGHJ6 at a significantly higher frequency (on average 39.3% in Kymice vs 24.5% in human repertoires), with reduced usage of IGHJ3, IGHJ4, and IGHJ5 (13.5% vs 8.5%, 45.2% vs 39.8%, and 12.8% vs 9.5% in human vs Kymouse repertoires) in Kymice (*Figure 2C*).

The genes encoding the $V_L$ likewise show a difference at the level of IGK/LV and IGK/LJ gene subgroup usage (*Figure 2D and E*) as is expected given the ~10% greater proportion of lambda chains found in Kymouse repertoires. Such differences in gene segment frequency persist when comparing frequencies within kappa and lambda chains (*Figure 2—figure supplement 1*). Within the kappa repertoire, there is significantly greater usage of IGKV1, IGKV2, and IGKV5 vs. IGKV3 in human repertoires (53.7% vs 46.7%, 16.8% vs 8.9%, 2.4% vs 0.1%, and 18.9% vs 35.3% in Kymice vs humans on average), as well as significantly greater IGKJ3, IGKJ4, and IGKJ5 usage and significantly reduced IGKJ2 usage (15.0% vs 10.7%, 29.3% vs 23.1%, 15.4% vs 8.6%, and 10.8% vs 26.2%). Within the lambda repertoire, Kymice use significantly less IGLV1 (20.1% vs 32.0%) and exhibited an increased usage of IGLV2, IGLV4, IGLV5, IGLV7, and IGLV9 (27.0% vs 20.9%, 3.2% vs 1.9%, 2.7% vs 0.8%, 3.6% vs 1.3%, and 3.0% vs 0.5%). There is also a significant reduction in the usage of IGLJ1 in Kymice vs. humans (8.4% vs 16.2%) and an increase in IGLJ3 usage (39.5% vs 31.1%).

At the level of gene subgroups, the Kymouse repertoires use a significantly more diverse set of light chain genes than do the human repertoires, when subsampling repertoires to the minimal sample size of 105 sequences (*Figure 2—figure supplement 2*). The Kymouse repertoires also use a significantly greater number of IGHD genes, but a significantly reduced number of IGHV genes. IGHV subgroup diversity is lower on average in Kymice but not significantly so. The reduced IGHV gene diversity and increased IGKLV gene diversity of the Kymouse result in comparable combinatorial diversity of these genes (*Figure 2—figure supplement 2*).

In order to elucidate whether the Kymice and human paired repertoires could be separated on the basis of their gene usage, we performed repeated random subsampling to the minimum repertoire size (105 sequences), calculated Z-normalised gene frequencies per repertoire, performed hierarchical

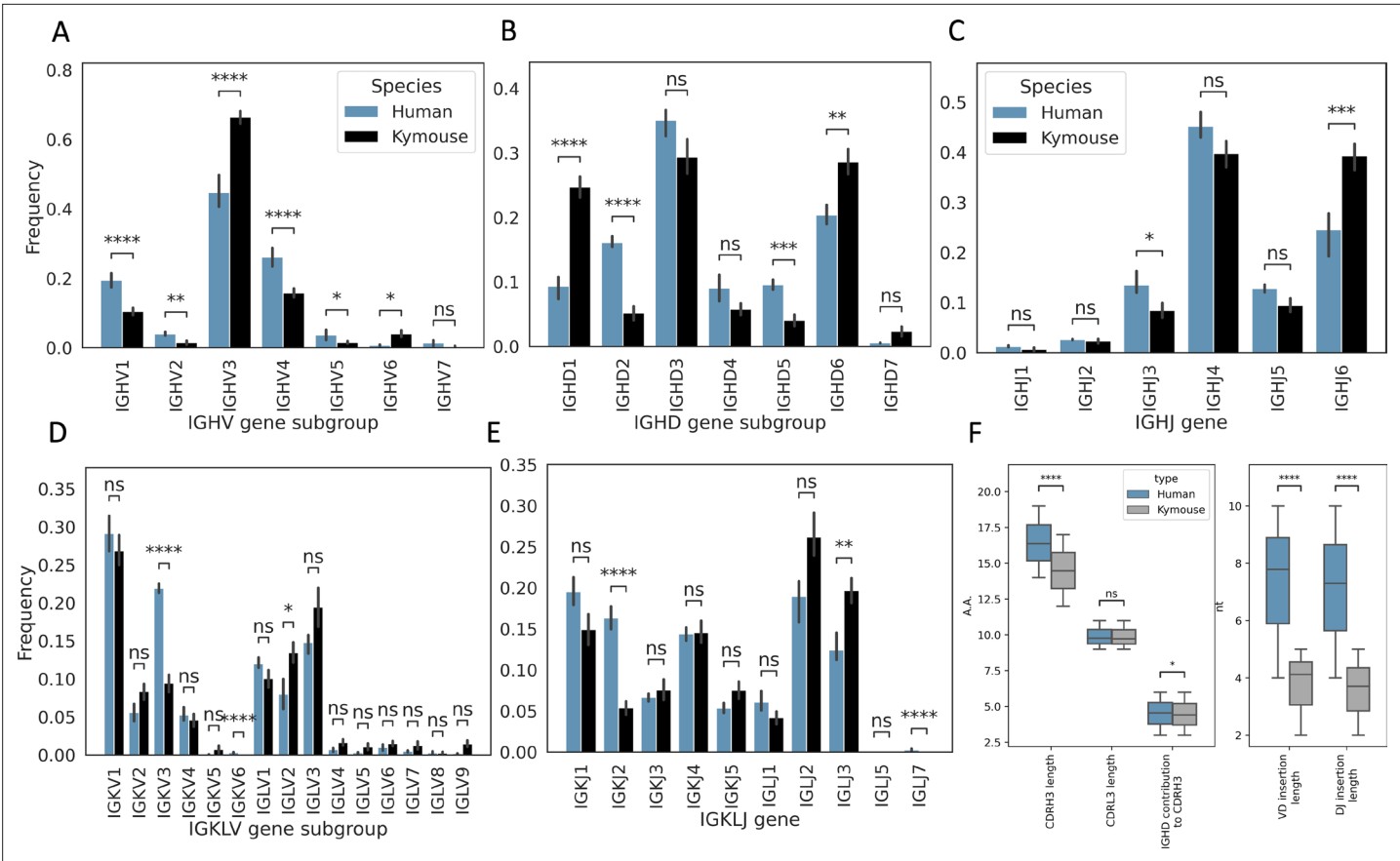

**Figure 2.** Single-cell sequencing of Kymouse BCR repertoires reveals significant differences in encoding gene frequencies in comparison to human repertoires. The non-mutated, naïve IgM subset of the paired $V_H$ and $V_L$ human and Kymouse sequences differ significantly in the frequencies of their encoding genes, as well as in their CDRH3 length, IGHD gene alignment length, and the length of their VD and DJ insertions. (**A**) shows the significant differences in the Kymouse's usage of IGHV gene subgroups with most notably significantly greater IGHV3 usage. IGHV3 comprised on average 66.4% of Kymouse repertoires vs 44.8% of human repertoires, and significantly lower IGHV1 usage (19.4% of human repertoires vs 10.5% of Kymouse repertoires). (**B**) likewise shows a number of differentially expressed gene subgroups: the largest differences are in IGHD1 (24.7% of Kymouse repertoires vs 9.3% of human repertoires on average), IGHD2 (16.1% of human repertoires vs 5.2% of Kymouse repertoires), and IGHD5 (9.6% vs 4.0% of Kymouse repertoires) and IGHD7 (28.6% of Kymouse repertoires vs. 20.4% of human). There are other notable differences in IGHJ gene usage (**C**) where there is significantly greater IGHJ6 usage in Kymice (39.3% vs 24.6% in humans) and significantly less IGHJ3 usage (8.5% vs 13.5% in humans). There are also differences in the genes encoding the light chain, the IGKLV (**D**) and IGKLJ (**E**) genes, such as significantly greater use of IGKV3 by human repertoires (21.9% of human repertoires vs 9.5% of Kymouse repertoires) and significantly greater use of IGLV2 by Kymouse repertoires (13.5% vs 8.0% of human repertoires on average). (**E**) shows significantly greater IGKJ2 usage by human repertoires (16.3% vs 5.4% in Kymouse) and significantly greater IGLJ3 usage by Kymouse repertoires (19.7% vs 12.4% in humans). (**F**) displays the distribution of CDRH3 length (which differs significantly with humans having CDRH3s on average 1.9aa shorter), CDRL3 length (no significant difference), and IGHD germline alignment length, which differs by 0.2aa on average, as well as the distribution of VD and DJ insertion lengths, which differ significantly and by nearly a factor of 2 (1.9× as many insertions on average in the VD junction, and 1.97 as many at the DJ junction).

The online version of this article includes the following figure supplement(s) for figure 2:

**Figure supplement 1.** Single-cell sequencing of Kymouse BCR repertoires reveals significant differences in encoding gene frequencies in comparison to human repertoires.

**Figure supplement 2.** Kymice repertoires appear to use a significantly more diverse set of IGHD genes (**A**), a comparable number of IGHJ genes (**B**), and significantly fewer IGHV genes (**C**) than human repertoires.

clustering and calculated the adjusted Rand index. The adjusted Rand index is a measure of how well the human and Kymouse repertoires could be clustered. A value of 1.0 indicates a perfect clustering. We found that the clearest separation could be achieved for IGK/LV gene usage, where Kymice and human repertoires could be perfectly separated in 96% of repeats. IGHD gene was the next best separator, clustering the repertoires separately in 62% of repeats. IGHV gene subgroup clustered

repertoires separately in 26% of repeats. IGHV gene, IGK/LV gene subgroup, IGK/LJ, and IGHJ gene clustered repertoires separately in 6%, 5%, 4%, and 0% of repeats separately.

We further examined the distributions of CDRH3 and CDRL3 length (*Figure 2F*). We found that human CDRH3s were on average longer than Kymouse CDRH3s (16.4aa ± 0.01 vs 14.5aa ± 0.13); there was no significant difference in CDRL3 length (average 9.8aa ± 0.002 vs 9.7aa ± 0.2). To identify possible causes for a difference in CDRH3 length, we compared IGHD gene alignment length and length of nucleotide insertions at the VD and DJ junctions. While there was a small but significant (5% level) difference in IGHD gene alignment length (average 4.6aa ± 0.06 vs 4.4aa ± 0.08), the largest difference is in insertion length at both the VD and DJ junctions (average 7.8nt ± 0.01 vs 4.1nt ± 0.1, 7.3nt ±0.02 vs 3.7nt ± 0.1). A significantly greater proportion of the Kymouse primary repertoire does not have junctional insertions compared to the human repertoire: 19.1% of the Kymouse sequences had no VD insertions, while 8.2% of the human sequences lacked VD insertions; 14.1% of Kymouse sequences had no DJ insertions while 4.5% of human sequences lacked these insertions. Over 10 times as many of the sequences in the Kymouse repertoires lacked junctional insertions compared to human repertoires (3.8% in Kymice vs. 0.3% in humans).

## Higher-depth bulk V$_H$ sequencing reveals differences in germline gene usage consistent with paired sequencing, and allows examination of CDRH3 diversity

While the single-cell data is a good measure of summary statistics such as gene usage frequencies, it is not sufficient to provide insight into features such as diversity within repertoires and overlap among repertoires, either at the level of clones or structurally. Bulk V$_H$ sequencing was used to increase the depth of our repertoire sampling by over 200-fold (from on average 218.4 heavy chain clonotypes per sample to an average of 53,006.2 per sample in the bulk V$_H$ experiments).

We used the bulk V$_H$ data to re-calculate gene usage frequencies in Kymice at a higher depth, and compare the usage frequency of the IGHV, IGHD, and IGHJ germline genes to those in bulk heavy chain human data of comparable depth. At such enhanced depth, all IGHD and IGHJ genes in the Kymouse haplotype were observed in all repertoires, as were all non-orphon IGHD and IGHJ genes in the general human database. There were 13 IGHV genes found in human repertoires that are absent in the Kymouse haplotype which were observed in some but not all human repertoires (*Figure 3—figure supplement 1*).

Statistical testing on a gene-by-gene basis was largely in agreement with the findings from the single-cell analysis (*Figure 3—figure supplement 2*) with the exception that IGHJ4 usage was elevated in the bulk data relative to the single-cell data, and IGHJ6 usage lower such that it is not significantly different from human repertoires (*Figure 3—figure supplement 3*).

In addition to statistical testing on a gene-by-gene basis (*Figure 3—figure supplement 2*), we used hierarchical clustering to compare the gene usage profiles of individual Kymice and humans, building dendrograms to show the relationships between the individuals' gene usage profiles. The frequencies as determined by sequence abundance are shown to differ in such a way that the two repertoire types can be clustered separately. The hierarchical clustering of the IGHV genes showed that the Kymice and humans form nearly separate monophyletic clusters except for a single outlier human subject (*Figure 3A*). Most of the variation in IGHV gene usage is explained by the IGHV gene subgroup usage: clustering by IGHV gene subgroup usage separates humans and Kymice without the outlier human sample, with Kymice using a lower proportion of IGHV1 and IGHV2 genes (5.3% vs 22.6% and 0.2% vs 2.1% in Kymouse vs human repertoires respectively for these subgroups), and an increased IGHV3 (51.6% vs 39.7%), IGHV4 (35.3% vs 31.6%), and IGHV6 usage (4.7% vs 1.0%) (*Figure 3B*).

The IGHJ gene usage profile is similar: Kymice and nine out of the ten humans form monophyletic clades with a single outlier human. On average, the Kymouse uses IGHJ4 more frequently than humans (46.7% vs 42.6%), and IGHJ5 and IGHJ1 less frequently (10.1% vs 13.5%, and 0.6% vs 1.0%) (*Figure 3C* and *Figure 3—figure supplement 3*). Both the IGHV and IGHJ gene usage profiles of naïve Kymouse repertoires are more similar to one another than to any human repertoire.

The IGHD gene usage is likewise distinguishable between humans and Kymice on average (*Figure 3D*). As can be seen from the heatmap, the IGHD germline genes used by the Kymice (e.g. IGHD3-22, IGHD2-15) are infrequently used by humans and vice versa. IGHD2, IGHD4, and IGHD5 subgroups are preferred by human repertoires (average 17.2% vs 5.8%, 9.1% vs 5.5%, and 9.5% vs

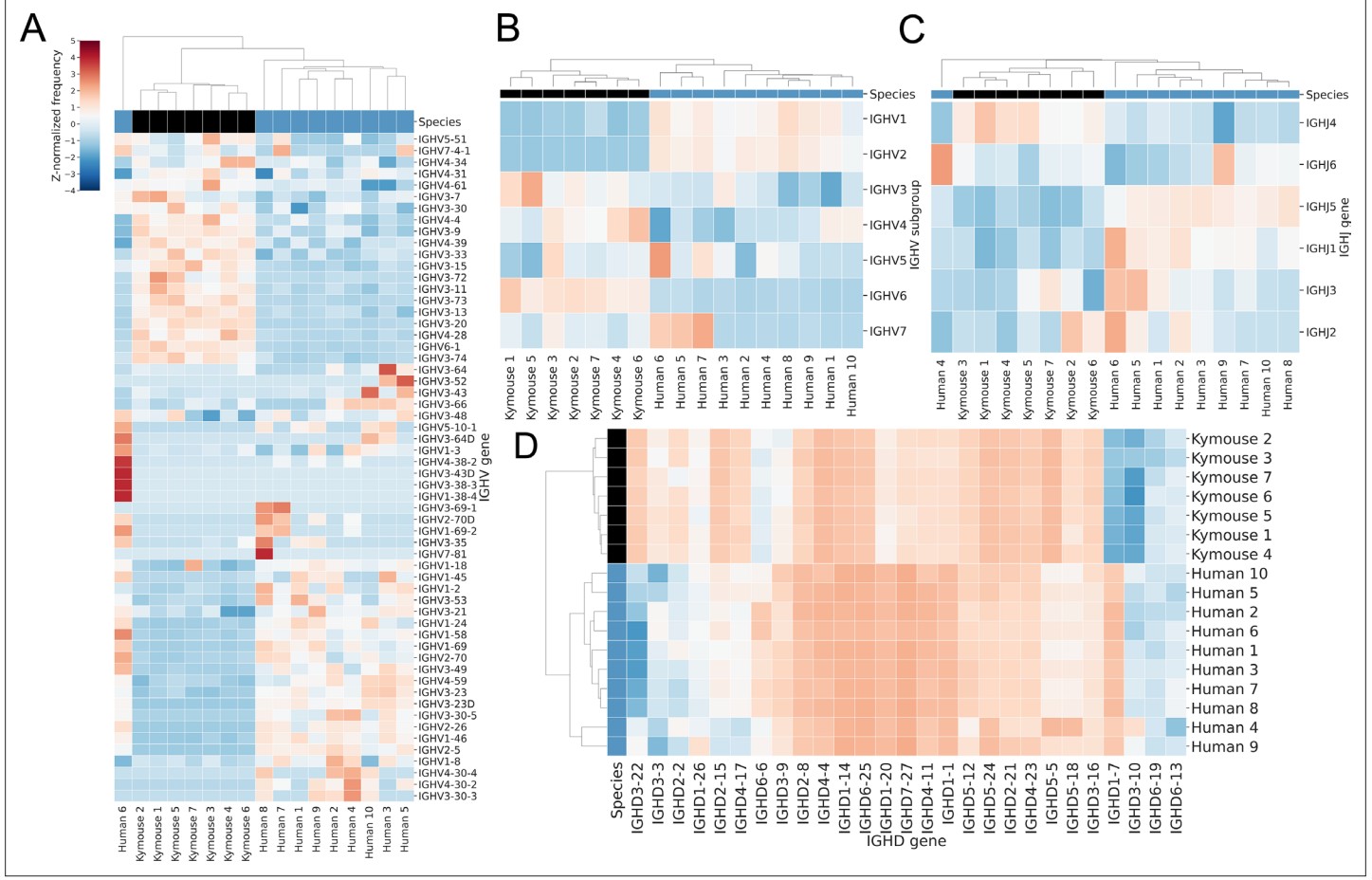

**Figure 3.** Bulk VH repertoires of Kymice exhibit differences in encoding gene frequencies versus human repertoires. Gene usage clustermaps for (**A**) IGHV genes, (**B**) IGHV subgroups, (**C**) IGHJ genes, and (**D**) IGHD genes from bulk V$_H$ sequencing reveal differences in gene frequencies that are sufficient to cluster most Kymouse and human repertoires. The IGHV clustermaps show a separation between human (blue) and Kymouse (black) repertoires, with lower usage of IGHV1 and IGHV2 in the Kymouse (5.3% vs 22.6% and 0.2% vs 2.1% in Kymice and humans, respectively) and increased usage of IGHV3 (51.6% vs 39.7%), IGHV4 (35.3% vs 31.6%), and IGHV6 (4.7% vs 1.0%). There are also differences in usage of IGHJ genes with a preference in the Kymouse repertoires for IGHJ4 (46.7% vs 42.6%). The IGHD gene usage shows the clearest distinction between Kymice and human repertoires with greater usage of IGHD1, IGHD6, and IGHD7 (24.2% vs 10.3%, 27.4% vs 20.4%, and 2.4% vs 0.6% on average in Kymice and humans, respectively) and lower usage of IGHD, IGHD4, and IGHD5 genes (17.2% vs 5.8%, 9.1% vs 5.5%, and 9.5% vs 4.8%) (*Figure 3—figure supplement 2*).

The online version of this article includes the following figure supplement(s) for figure 3:

**Figure supplement 1.** The Kymouse repertoires use significantly fewer IGHD (**A**) and IGHV (**B**) genes than do the human repertoires: this is because the 'missing' IGHV and IGHD genes are absent from the inserted haplotype in the Kymouse locus.

**Figure supplement 2.** Kymice have on average significantly different IGHV, IGHD, and IGHJ gene usages to humans in their primary repertoires.

**Figure supplement 3.** Comparison of IGHJ usage between naïve human and Kymouse IgM B-cell receptor (BCR) repertoires (corresponding to subplot C of *Figure 3—figure supplement 2*).

4.8% respectively in human vs Kymouse repertoires), while Kymouse repertoires preferentially use IGHD1, IGHD6, and IGHD7 (24.2% vs 10.3%, 27.4% vs 20.4%, and 2.4% vs 0.6% on average in Kymice and humans, respectively) (*Figure 3—figure supplement 2*).

With the greater depth afforded by bulk V$_H$ sequencing, we were able to analyse further the differences in CDRH3 length we noted in the single-cell analysis. In addition to our comparison with naïve human repertoires, we compared the Kymouse repertoires with equivalent repertoires of C57BL/6 mice (non-mutated IgM sequences from naïve B cells). We compared the distribution of the CDRH3 lengths in each species' naïve repertoire (*Figure 4A*). This revealed that Kymice have an average CDRH3 length in between that of humans and mice, with a mean CDRH3 length of 14.3 amino acids. In comparison, the C57BL/6 mouse dataset has a mean length of 12.4 amino acids, and the human

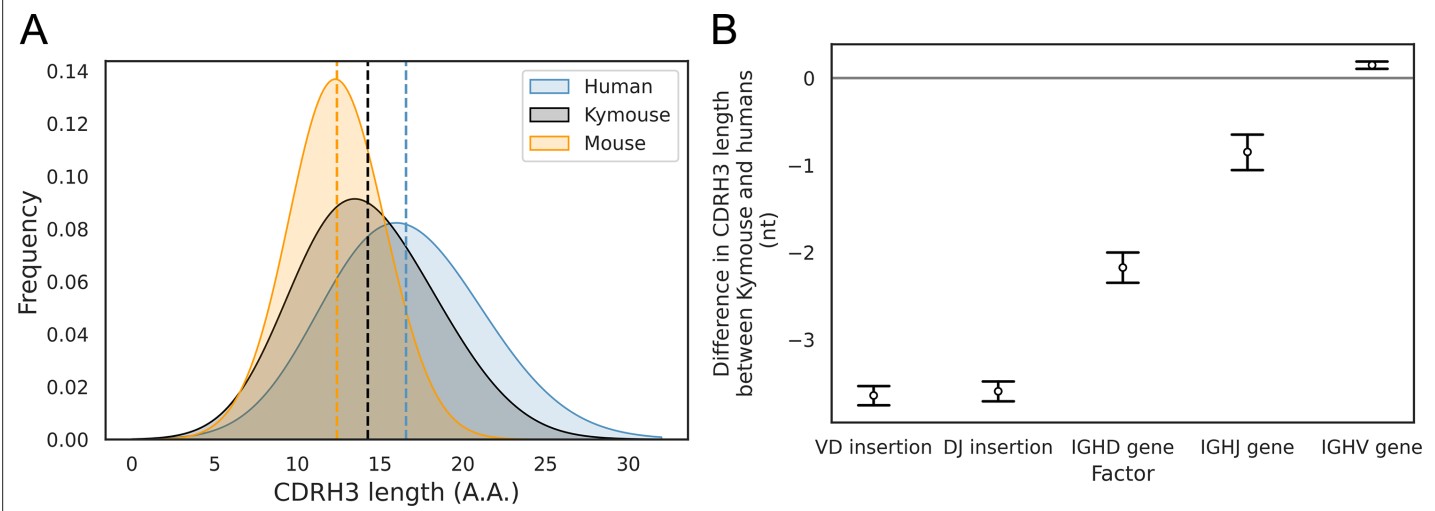

**Figure 4.** The CDRH3 length distribution of the Kymouse is intermediate between humans and C56BL/6 mice primarily due to reduced VD and DJ insertion rates. The CDRH3 length distribution of the Kymouse (average 14.3aa) is intermediate between equivalent C57BL/6 repertoires (12.4aa) and human repertoires (16.6aa) (**A**). For each of five possible contributing factors, we used bootstrapping to estimate the contribution of the factor to difference in CDRH3 length, measured in nucleotides. Where this factor is positively contribution (i.e. is above zero), the factor is leading to differentially longer CDRH3s in the Kymouse; where negative, it leads to differentially shorter CDRH3s in the Kymouse. These statistics reveal that the major factor leading to a reduction in CDRH3 length in the Kymouse, despite access to the same germline repertoire, is the relative lack of VD and DJ insertions (**B**) (**Figure 4—figure supplement 2**).

The online version of this article includes the following figure supplement(s) for figure 4:

**Figure supplement 1.** The distribution of average CDRH3 length per repertoire is shown for each of the bulk VH repertoires.

**Figure supplement 2.** VD and DJ insertion length distributions in the Kymouse (grey) versus human (blue) repertoires.

dataset has a mean CDRH3 length of 16.6 amino acids. Kymouse CDRH3 loops are on average 2.36aa shorter than humans (95% CI: 2.26, 2.48; p<0.001), while C57BL/6 mice CDRH3 loops are on average 4.21aa shorter than humans (95% CI: 4.12, 4.30; p<0.001). The intra-species variance in CDRH3 length is small in comparison to the inter-species difference defined above (**Figure 4—figure supplement 1**).

To establish the underlying mechanism of the CDR3 length difference, we first looked at whether the differential IGHV gene usage between humans and Kymice had a significant impact on the length of the CDRH3 loop. For each $V_H$ sequence in the human and Kymouse datasets, we determined the number of nucleotides that the IGHV germline gene contributes to the CDRH3 loop. The results showed a statistically significant but small difference of 0.14nt (95% CI: 0.12, 0.16; p<0.001) between humans and Kymouse. Therefore, it does not appear that the differential choice of IGHV between humans and Kymouse greatly affects the CDRH3 length. We next investigated the effect of the differential usage of IGHD genes between humans and Kymice on the length of the CDRH3 loop. This showed that the human IGHD germline genes used by the Kymouse are, on average, 2.34nt shorter than humans (95% CI: 2.26, 2.41; p<0.001). We then compared the relative usage of the IGHJ genes of humans and Kymice (**Figure 3—figure supplement 3**). Kymice tended to use IGHJ4 (47.3% in Kymice versus 44.2% in humans) and IGHJ6 (32.8% in Kymice versus 28.6% in humans) while using the other genes (IGHJ1, IGHJ2, IGHJ3, IGHJ5) slightly less frequently, in particular IGHJ5 (11.1% in Kymice versus 15.1% in humans). Estimation statistics revealed that the differential IGHJ gene usage between Kymouse and human resulted in a decrease in the CDRH3 length for Kymouse of 0.85nt (95% CI: 0.77, 0.94; p<0.001).

Finally, we looked at the number of nucleotide insertions in the VD and DJ junctions. The results showed that Kymice VD junctions are on average 3.73nt shorter than humans (95% CI: 3.68, 3.78; p<0.001), with a mean insertion size of 3.35nt compared to 7.30nt for humans (**Figure 4—figure supplement 2**). Equally, the Kymice DJ junctions are on average 3.68nt shorter than humans (95% CI: 3.63, 3.73; p<0.001), with a mean insertion size of 2.91nt compared to 6.77nt for humans (**Figure 4—figure supplement 2**). Overall, the number of junctional nucleotides inserted in the Kymouse heavy chain is 7.33 fewer than in humans (95% CI: 7.25, 7.40; p<0.001), with an average of 7.55 in Kymice

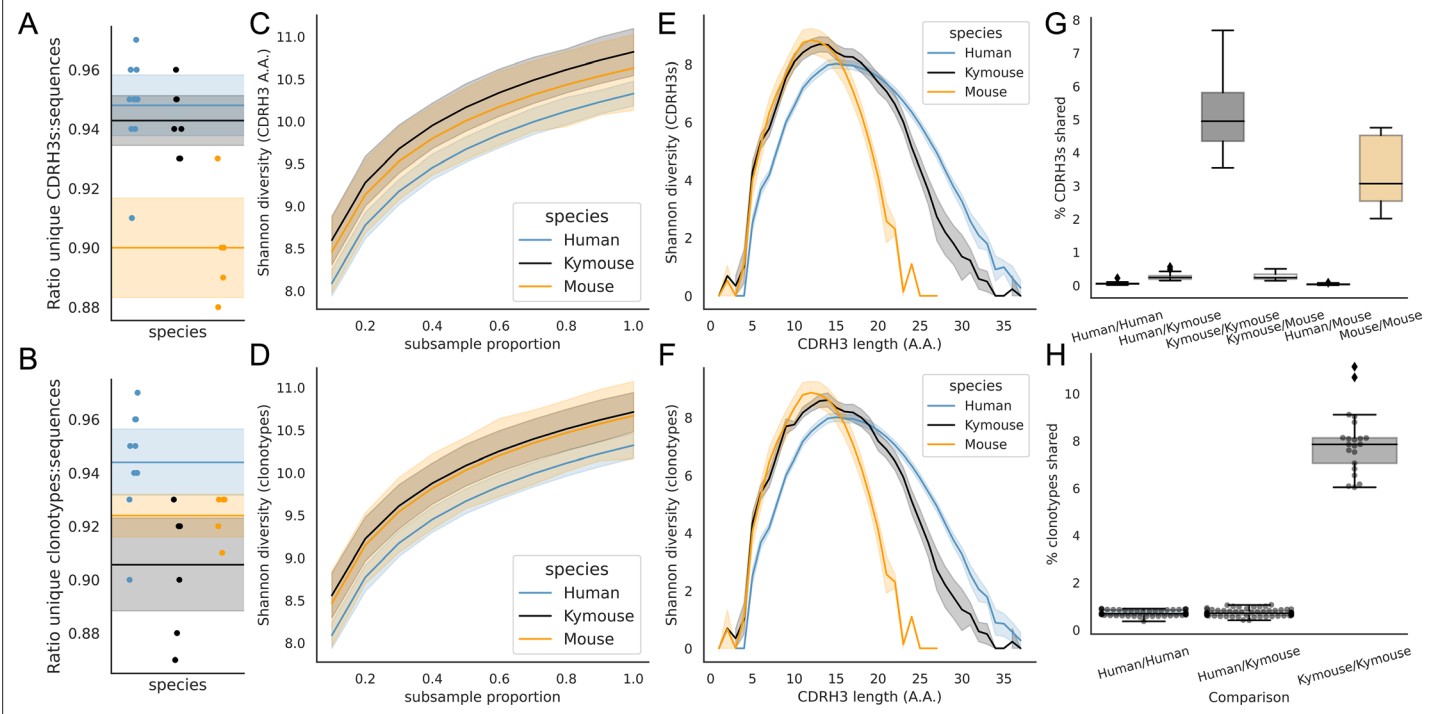

**Figure 5.** Examination of the CDRH3 and clonal diversity in the primary B-cell receptor (BCR) repertoires of humans, Kymice, and mice reveal comparable diversity on a repertoire level, despite the described differences in CDRH3 lengths. Overlap analysis reveals considerably greater diversity among human repertoires. The top rows pertain to exact CDRH3 (amino acids) and the bottom rows to clonotypes (same IGHV, IGHJ, and greater than 90% amino acid identity across length-matched CDRH3s). At the level of CDRH3s, the Kymouse repertoires have more unique CDRH3s per sequence sampled (**A**), are more diverse in their usages (**C**), despite their limited VD and DJ insertion rates. Diversity is reduced relative to human sequences at longer CDRH3 lengths which in unmutated repertoires require VD/DJ insertions to reach (**E**). Kymouse repertoires show an opposite pattern in unique sampling rate and diversity when looking at clonotypes (**B and D**, respectively) but still show reduced diversity versus humans at longer lengths. Overlap among CDRH3s (**G**) and clonotypes (**H**) between individuals is considerably higher in Kymice than humans, and more comparable to mice (**G**).

compared to 14.88 in humans. These results show that the main factors that give rise to the shorter CDRH3 lengths in the Kymouse compared to humans is the reduced number of nucleotide insertions in the VD and DJ junctional regions, and the differential usage of IGHD germline genes between the species (*Figure 4B*). This is consistent with our results from the single-cell analysis.

A correlate of shorter CDRH3 length is a reduction in the theoretical CDRH3 diversity obtainable, particularly given that this is caused by a reduction in the non-templated insertion of nucleotides. To examine the diversity of the repertoires, we considered how many unique CDRH3s or clonotypes can be found in a given repertoire, normalised by the total number of sequences observed. *Figure 5A* shows that the ratio of the number of unique CDRH3s to the number of sequences is comparable in Kymice and humans, and that both are considerably higher than in C57BL/6 mice.

When clustering CDRH3s with 90% amino acid identity in combination with common IGHV and IGHJ genes (clonotypes), there are comparable unique clonotypes per sequence in mice, humans, and the Kymouse (*Figure 5B*). We calculated the Shannon diversity of the CDRH3 and clone distributions, a measure which takes into account both richness and abundance and found comparable diversity across all repertoire types (*Figure 5C and D*). The diversity observed depends on CDRH3 length considered, with the mouse and Kymouse repertoires having greater diversity at shorter lengths (peak diversity at lengths 12 and 13, respectively) and the human repertoires having greater diversity at lengths greater than 18 amino acids (peak diversity at length 15) for both CDRH3s (*Figure 5E*) and clonotypes (*Figure 5F*). Given that junctional insertions are required to reach lengths of 23 or greater in the human and Kymouse data, this supports the hypothesis that the reduced junctional diversification in naïve Kymouse repertoires limits CDRH3 diversity at the longest lengths, however the greater diversity at shorter lengths is seen. It is also clear that the Kymouse occupies a region in CDRH3 diversity and length between wild-type mice and humans.

Finally, we considered the CDRH3 and clonotype overlap among mice, Kymice, and humans (*Figure 5G and H*). The overlap among CDRH3s is highest between individual mice and individual Kymice. On average, 5.1% of CDRH3s were shared between any two Kymice and 3.4% of CDRH3s shared between any two mice, compared to just 0.12% of CDRH3s shared between any two human subjects. Despite comparable CDRH3 diversity reported for individual Kymice and humans, over 25 times more CDRH3s are shared between pairs of Kymice on average than between humans. Clonotype sharing was also higher between individual Kymice (average 7.92% vs 0.84% in humans). Interestingly, clonotype sharing was significantly higher among individual Kymice (average: 7.92%, range: 6.04–11.14%) than between individual mice (average 4.55%, range: 3.52–5.90%) despite comparable CDRH3 sharing.

## The Kymouse repertoires are structurally more human-like than mouse-like

Ultimately, genetic diversity is reflected in the structure of the resulting BCR and secreted antibody protein. Therefore, we compared the structural similarities of the BCR repertoires via structural annotation of the CDRs. We first compared the CDRH1 and CDRH2 loops, which adopt a limited set of conformations known as canonical classes. For both CDRH1 and CDRH2 the Kymouse repertoires group separately from human repertoires, but group with human repertoires before the mouse repertoires, in their usage of canonical forms (*Figure 6—figure supplement 1*). All canonical forms are strongly predicted by IGHV germline gene subgroup (*Supplementary file 1*) especially as these sequences are non-mutated. Despite the effects of the differential IGHV usage observed in humans and Kymice, this does not make the Kymouse canonical class usage more similar to mice as the murine repertoires use different canonical classes than either the human or humanised repertoires. Interestingly, six of the nine murine IGHV germlines encode a single canonical form, H1-8-A, suggesting more limited structural diversity in murine CDRH1s.

We then performed structural comparisons of the shapes of the human, Kymouse, and mouse CDRH3s. CDRH3s do not adopt canonical conformations, so we used two different approaches: firstly, structural annotation which consists of comparison to a CDRH3 structural database and annotation with a structural cluster ID, and secondly full CDRH3 modelling. Of the 1944 possible structural clusters in the CDRH3 structural database, 1594 were observed in at least one repertoire. The majority (1270) of these clusters were observed in all species (*Figure 6—figure supplement 5*). There was an observable difference among repertoires in the species origin of the structural clusters observed, i.e., each species was biased in the structural space it tended to use. The majority of structural clusters used in human and Kymouse repertoires were of human origin (57.5% and 55.0%, respectively), while the majority of structural clusters used in mouse repertoires were of murine origin (64.0%) (*Figure 6—figure supplement 2*).

As described, the CDRH3 lengths in the human dataset were on average 2.36aa longer than those in the Kymouse, and 4.21aa longer than those in the C57BL/6 mice (*Figure 4A*). As we considered only CDRH3s between length 4 and 16 in this structural analysis, this difference was reduced to a difference of 0.45 amino acids between human and Kymouse repertoires (CI 0.43, 0.47) and 0.63 between human and mouse repertoires (CI 0.62, 0.65). The average CDRH3 lengths of the human templates in the FREAD database were longer than those in the murine templates (12.6aa compared to 11.2aa, respectively) (*Figure 6—figure supplement 3*). We checked that the observed differences in structural template usage by humans, Kymice, and C57BL/6 mice were not just reflecting differences in the availability of templates at the difference CDRH3 lengths. Therefore, we stratified the FREAD database by CDRH3 length and ran the datasets against each CDRH3 length separately. If the preferential usage of human-specific templates by humans and humanised mice, and vice versa for mice, is simply a result of random sampling from the FREAD database, we expect the ratio of the proportion of human templates in human repertoires to the proportion of human templates in the FREAD database to be approximately 1 across all CDRH3 lengths, and below 1 across all CDRH3 lengths for mice. Instead, we see a significant enrichment (at the 1% level) of species-specific templates at multiple CDRH3 lengths.

The human repertoires are more structurally variable than are the humanised murine or murine repertoires (*Figure 6A, B and C*). The murine repertoires are the least variable (there is the smallest range of distances between any two given subjects for mice, and on average the smallest average

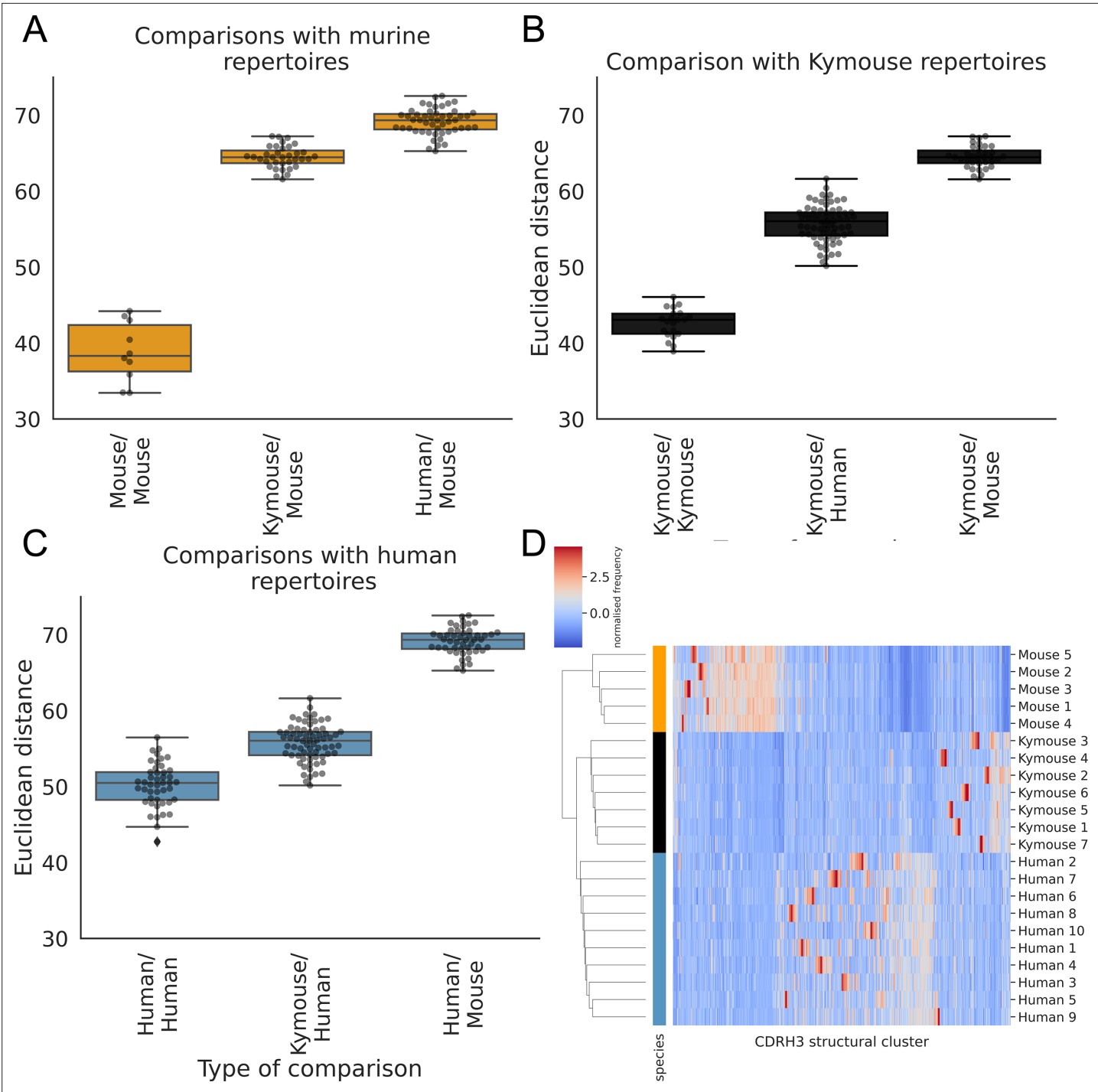

**Figure 6.** The CDRH3 structural repertoire of Kymice is human-like: distance between CDRH3 structural cluster usage profiles is measured with Euclidean distance in Z-normalised proportions. This is calculated pairwise between subjects and these distances are clustered hierarchically. **A through C** show these pairwise distances stratified by the type of comparison. The leftmost box shows the range in distance between individuals of the same species. For the mouse and Kymouse repertoires, this range is smaller than the range in distances for any other species, meaning that they cluster monophyletically (**D**). Human repertoires have less self-similar CDRH3 structural cluster usages with ranges overlapping with Kymouse repertoires. In the hierarchical clustering solution with these pairwise distances that is shown in (**D**), the human and Kymouse repertoires form a monophyletic clade separately from the murine repertoires.

The online version of this article includes the following figure supplement(s) for figure 6:

**Figure supplement 1.** CDRH1 (**A and C**) and CDRH2 (**B and D**) canonical class usage clusters Kymice and humans separately from mice.

*Figure 6 continued on next page*

*Figure 6 continued*

**Figure supplement 2.** The majority of CDRH3 structural clusters are observed across all three repertoire types.

**Figure supplement 3.** Around 40.5% of templates in the FREAD database are labelled as of murine-origin vs. 37.1% of human templates.

**Figure supplement 4.** Structural clustering of CDRH3s from antibody models produced by ABlooper indicate greater structural similarity between the repertoires of humans and Kymice than to C57BL/6 mice, as per the homology-based annotation pipeline.

**Figure supplement 5.** The majority of CDRH3 structural clusters are shared among repertoires.

distance between pairs of subjects). All the Kymouse repertoires were structurally closer (Euclidean distance in Z-normalised CDRH3 structural cluster usage) to any given human repertoire than to any C57BL/6 mouse repertoire (*Figure 6A, B and C*). With no correction for sample size or for CDRH3 length distribution, the humanised murine and human repertoires form a monophyletic cluster that is sister group to the murine repertoires (*Figure 6D*).

Labelling cluster members as 'murine' or 'human-like', the adjusted Rand index of this clustering is 1.0 (perfect correspondence). We calculated Rand indices for 100 subsamples with equal numbers of each CDRH3 length to control for any length effect; in all subsamples, the human and humanised repertoires could be clustered separately from the murine repertoires resulting in an adjusted Rand index of 1.0. Further, in all 10 subsamples, at least one human repertoire was more structurally similar to a given humanised murine repertoire than to at least one other human repertoire (between 5 and 9 of 10 subjects per subsample). In conclusion, even when adjusting for sample size and differing CDRH3 length distribution, the repertoires of Kymouse are structurally more similar to human repertoires than they are to murine repertoires according to the homology-based structural annotation technique.

The species bias we observed in structural cluster usage, which is assigned via sequence similarity to a structural template, meant that a template-free modelling procedure might lead to different results. We modelled non-singleton clonotype representatives from humans, Kymice, and mice using the deep learning tool ABlooper, then compared the resultant CDRH3s via $C_\alpha$ RMSD. Analogously to clustering of the CDRH3 template database, we performed greedy clustering with a 0.6 Å cut-off. This clustered the 43,378 models with 41,397 unique CDRH3s into 6546 clusters. The modelled human CDRH3s were on average longer than Kymouse CDRH3s by 0.99 amino acids (CI: 0.94, 1.05) and longer than mouse CDRH3s by 1.23 amino acids (CI: 1.17, 1.29). We observed a slightly different clustering by usage than via the homology approach (*Figure 6—figure supplement 4A*), with monophyly of Kymice, and Kymice and humans, but not of mice. The distribution of intersubject differences was such that the human/Kymouse and human/human comparison distributions are largely overlapping. Indeed, the differences between all distributions were less extreme with overlap between most of the distributions (*Figure 6—figure supplement 4B*). The monophyly of Kymice and humans versus mice is observed when subsampling in order to equalise the CDRH3 length distribution, i.e., it is not driven by differences in CDRH3 length. The deep learning-based modelling approach supports the earlier finding that Kymice CDRH3 repertoires are more structurally similar to human CDRH3 repertoires than murine repertoires, and vice versa; the extent of this similarity is greater than observed with the homology modelling approach.

In conclusion, the sequence differences in the repertoires which were described in the germline and diversity sections do impact upon the repertoire of CDR structures which are observed in the Kymouse. The distribution of canonical forms in CDRH1 and CDRH2 is distinguishable, but usage is human-like. In the CDRH3, both structural annotation and full structure prediction indicate that the naïve Kymouse CDRH3 structural space is human-like, indicating that Kymice repertoires offer comparable structural starting points for the production of antigen-specific antibodies.

## A state-of-the art humanisation tool scores Kymouse sequences as fully human

We finally tested whether the Kymouse sequences are considered human by a state-of-the-art humanisation tool. We used the random forest (RF) classifiers within Hu-mAb to score heavy chain amino acid sequence humanness. One-hundred percent of human and Kymouse heavy chain sequences were classified as human, with the maximum humanness score assigned to 99.1% of human sequences and 98.3% of Kymouse sequences. All sequences produced scores in the 'Positive (High Score)' category

which had minimal anti-drug antibody events reported among therapeutic antibodies. No murine sequences were classified as human.

## Discussion

The phenotypic diversity of B cells in the spleens, lymph nodes, and bone marrows of the Kymouse, determined by immunophenotyping panels, showed that the main immunologically relevant B-cell subpopulations could be identified at appropriate cell frequencies, consistent with the Kymouse being fully competent for B-cell development and capable of a complete humoral immune response. As expected, the baseline levels of the immune relevant subsets, i.e., memory B cells and GC B cells in the spleen and lymph nodes as well as the plasma cells in bone marrow were low reflecting the lack of immune exposure beyond commensal and environmental antigens during mouse husbandry.

While somatic hypermutation plays a key role in the development of mature high-affinity antibodies, the breadth of an immune response to an antigen is limited at first by the diversity produced by recombination and non-template additions of the non-somatically hypermutated, germline-encoded heavy chain Ig genes and their pairing with similarly rearranged Ig light chain genes that together comprise the BCRs in the naïve B cells. Paired $V_H/V_L$ sequencing of the naïve Kymouse BCR repertoire revealed a near 50:50 Igκ/Ig $\lambda$ ratio which more closely approximates the human Igκ/Ig $\lambda$ ratio of 60:40 than the murine Igκ/Ig $\lambda$ ratio of 95:5. Analysis of encoding gene frequencies revealed different average gene frequencies in all encoding gene categories between Kymice and humans: IGHV and IGKLV genes and gene subgroups, IGHD gene subgroups and IGHJ genes. There was also a significant difference in CDRH3 length with significantly shorter CDRH3s in the Kymouse repertoires than in human repertoires, in part due to IGHD gene usage with decreases in the length of the IGHD gene in the CDRH3. However, deep unpaired $V_H$ sequencing which increased our sequence depth by greater than 200-fold showed that naïve Kymouse BCRs have a CDRH3 length distribution that is intermediate between human and mouse repertoires (on average ~14aa for Kymouse versus ~16aa for human repertoires and ~12aa for mice). A great proportion of this effect is ascribable to fewer nucleotide insertions at both the VD and DJ junctions during junctional diversification in the Kymouse with on average 7.33nt fewer inserted in Kymouse compared to humans over the two junctions. This reduced junctional diversification in the Kymouse leads to lower diversity in longer CDRH3s and greater clonotype overlap between individual Kymice than between individual humans, consistent with the observation that junctional insertions are a key factor in the publicness of antibody CDRH3s (*Jaffe et al., 2022*).

Deep unpaired $V_H$ sequencing also showed that the IGHV, IGHD, and IGHJ gene usage profiles in Kymice are distinct from human profiles consistent with our lower depth paired analysis. However, IGHV and IGHJ gene usage frequencies did not perfectly separate Kymice from human repertoires in clustering analyses, indicating that these Kymice gene usage frequencies might fall within natural human variation in these frequencies. This contrasts with the previous NGS characterisation of the OmniRat in which both IGHV and IGHD gene usage was distinct from humans (*Joyce et al., 2020*). However, Kymouse repertoires did show a preference for shorter IGHD genes: this was also observed in the NGS characterisation of OmniRat BCR repertoires, suggesting that a preference for shorter IGHD genes may be common across transgenic rodent platforms (*Joyce et al., 2020*). Nevertheless, different germline gene distributions in the Kymouse may affect how representative transgenic models are of immune responses to germline-targeting immunogens especially if a desired naïve BCR frequency is at lower level than in humans (*Chen et al., 2019*). Data present here allows the pre-assessment of gene usage frequencies to interpret such studies.

While antibody sequence diversity is clearly the primary determinant of the functional diversity of a repertoire, there exists some sequence and structural redundancy. We decided therefore to look at whether there were differences between repertoires in terms of their distribution of CDRH3 structures: changes in CDRH3 structural cluster usage have been previously observed along the B-cell development axis (*Kovaltsuk et al., 2020*) and allow comparison of repertoires derived from different species. Despite the differences in germline gene usage established here, modelled structural comparison of the human and humanised repertoires to the murine repertoires showed murine repertoires use mostly CDRH3 structural clusters that have been identified from murine antibodies, while human and Kymouse repertoires use CDRH3 structural clusters identified from a more even distribution of species of which more than 50% were identified from human antibodies. Further, grouping of the exact distribution of CDRH3 structural clusters reveals that Kymouse structural

cluster usage is closer to human usage than murine usage, accounting for CDRH3 length differences. When modelling the CDRH3s (as opposed to performing an approximation via structural annotation), part of the 'structural distance' between the human, mouse, and Kymouse repertoires disappeared suggesting part of the signal observed in homology modelling is due to different sequences with similar predicted shapes. This suggests that despite the observed differences at the sequence level, the CDRH3 structural shapes adopted by the Kymice BCRs are within the distribution of observed human shapes.Finally, using the Hu-mAb humanness classifiers, all Kymouse and human sequences are classified as human, meaning that naïve sequences isolated from the Kymouse are predicted to have similar immunogenicity in humans to sequences isolated from humans themselves.

Kymice contain the Ig haplotype of a single human subject consistent with reduced diversity relative to outbred human B-cell repertoires where the considerable allelic and haplotypic diversity observed in human populations has significant effects on the expressed BCR repertoire (*Rodriguez et al., 2022*). Given the Kymouse's use of a single haplotype, and differences in the observed germline gene usages, it is likely that the BCR response to antigen stimulus in the Kymouse will represent only a subset of the possible diversity of the outbred human response and this should be in consideration in vaccine response modelling.

In conclusion, although naïve BCR repertoires of the Kymouse have key distinctions from human repertoires at the sequence level they are comparable to the human repertoires in terms of CDRH3 structural usage. The less diverse primary repertoire of the Kymouse does not appear to be an impediment to its use as a source of high-affinity therapeutic antibodies, and indeed a higher frequency of shorter CDRH3 lengths could be favourable in terms of the developability profile of any mAbs derived from these systems (*Raybould et al., 2019*). In addition to the analysis of naïve, antigen-inexperienced repertoires in the current study, we and others have also used the Kymouse as a model system in multiple immunisations with a broad range of different viral, bacterial, and parasite antigen targets, as well as self-antigens for therapeutic applications (*Walls et al., 2020*; *Jiao et al., 2020*; *Sainson et al., 2020*; *Petzer et al., 2020*; *Hastie et al., 2021*; *Richardson et al., 2021*; *Saghari et al., 2022*). The polyclonal serum response elicited in these Kymice has been shown to be functional and the isolated monoclonal antibodies bind physiologically important epitopes recognised by similar human antibodies (*Rujas et al., 2020*). Indeed, a number of studies have showed that the Kymouse is able to elicit monoclonal antibodies equivalent to those found in humans exposed to the same antigen (*Sok et al., 2016*; *Scally et al., 2017*; *McLeod et al., 2019*; *Oyen et al., 2020*). Together these data suggest that the engagement of Kymouse BCRs on naïve B cells is authentic, and that the structural templates available for antigen binding are indeed human-like.

## Materials and methods

### Kymouse data

Spleens, lymph nodes, and bone marrows were collected from 22 antigen-inexperienced Intelliselect Transgenic mice (Kymice). These Kymice contain chimeric Ig loci, with humanised variable domains ($V_H$, $V_K$, and $V_L$) and a humanised lambda constant domain ($C_L$), but murine heavy ($C_H$) and kappa ($C_K$) constant domains (*Lee et al., 2014*). The Kymice were selected to reduce any possible confounding effects by ensuring that: (i) there was an equal representation of sexes, (ii) mice were a range of ages on culling (6–12 weeks of age), and (iii) mice were selected from different litters and culled over a period of 2 years. Twelve of these Kymice were used for the B-cell phenotypic characterisation. The number of Kymice used in each experiment is shown in *Figure 7*.

### Lymphoid cell isolation and cryopreservation

Bone marrow isolated from the femurs and tibias of each Kymouse were processed to single-cell suspensions by flushing the tissues with ice-cold FBS buffer and passing through 40 µm cell strainers. Spleens and inguinal lymph nodes were processed to single-cell suspensions by homogenising through 40 µm cell strainers with ice-cold FBS buffer and pooled prior to staining and cell sorting. All single-cell suspensions were pelleted at 400 × *g* for 10 min at 4°C prior to cryopreservation in 10% DMSO/FBS and storage in liquid nitrogen.

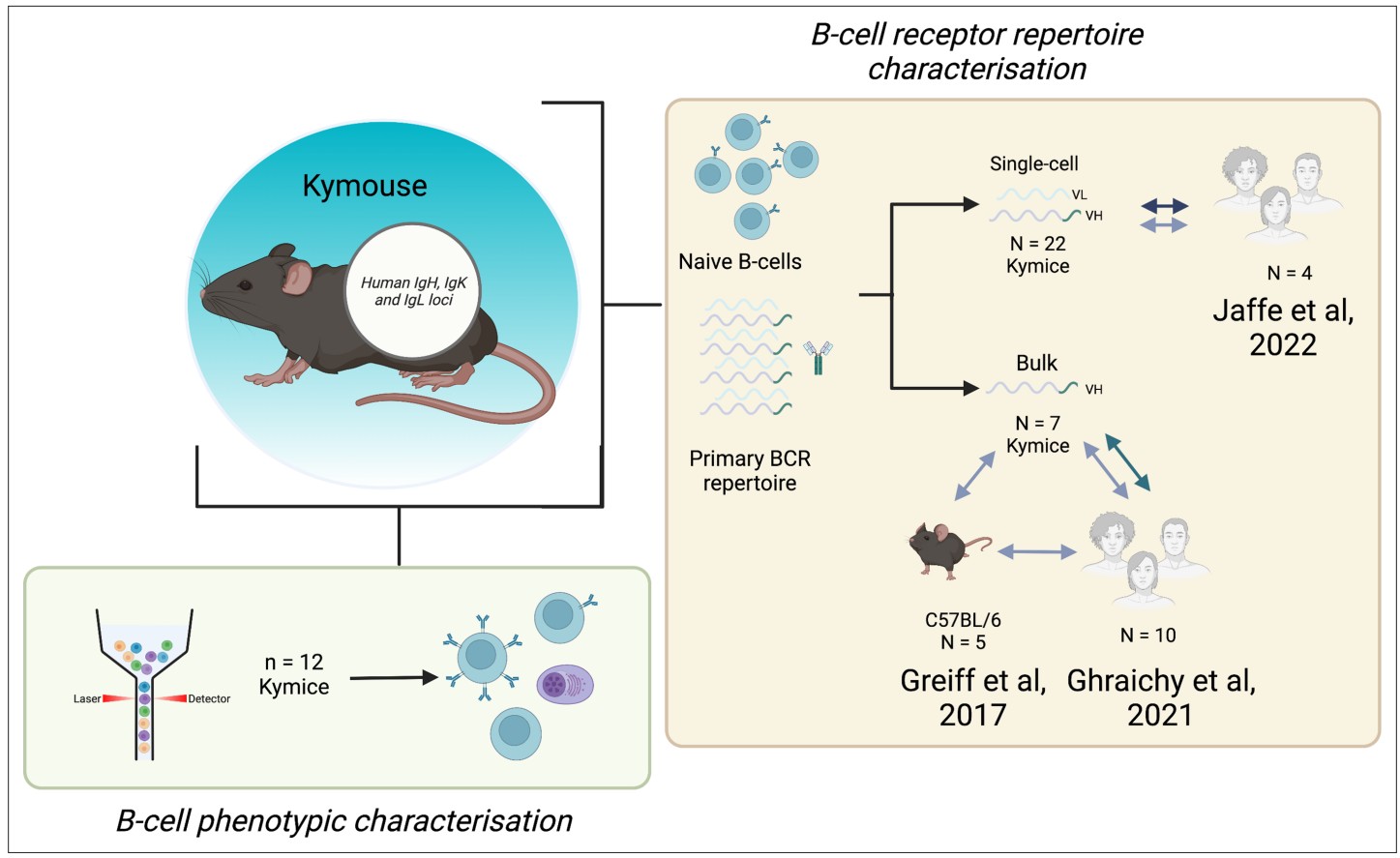

**Figure 7.** Experimental overview. The bone marrow, spleen, and inguinal lymph nodes of 22 Kymice were used variously for B-cell phenotypic characterisation (bone marrow) and B-cell receptor (BCR) repertoire characterisation at both the single-cell paired $V_H$/$V_L$ (spleen and lymph nodes) and bulk $V_H$ level (bone marrow).Single-cell $V_H$/$V_L$ data and bulk $V_H$ data were analysed in comparison to previously published naïve datasets from C57BL/6 mice and humans.

## NGS analysis of paired $V_H$ and $V_L$ sequences from single-cell sorted B cells derived from Kymice

For 22 Kymice, the spleen and inguinal lymph nodes were processed to single-cell suspensions as described above before FACS to recover CD19[+] B220[+] B cells into individual wells of a 96-well plate. RT-PCR was performed to amplify the $V_H$ and $V_L$ domains, and standard 2×300 bp Illumina libraries were generated before sequencing on an Illumina MiSeq sequencer as previously described (*Lee et al., 2014*). Unique molecular identifiers (UMIs) were used to group derivatives of the same RNA molecules.

pRESTO was used to process the reads and build UMI consensus groups (*Vander Heiden et al., 2014*). The Change-O pipeline (*Gupta et al., 2015*) was then used to process the resultant sequences; naïve BCR sequences were characterised as IgM sequences containing zero nucleotide mutations. In total, 3175 paired $V_H$ and $V_L$ sequences were processed from the 22 Kymice. In subsequent analyses comparing gene frequencies, we removed data from seven Kymice where fewer than 100 paired sequences were retrieved, resulting in 15 Kymice contributing to the final dataset. In these 15 Kymice, the average sample size was 202 (range: 105–363).

## NGS sequence analysis of published paired $V_H$ and $V_L$ sequences from naïve human B cells

As a human comparator for the single-cell Kymouse data, we used a recently published, high-depth paired sequencing dataset, selecting the subset of IgM sequences with zero mutations from naïve B cells as made available by *Jaffe et al., 2022*. We removed sequences from cells in which there were

multiple heavy or light chains. This resulted in 478,177 unique total sequences across four subjects (average: 119,544 sequences per subject; range: 88,631–143,293).

## NGS sequence analysis of V$_H$ sequences from bulk sorted B cells derived from Kymice

Bone marrows from the femur and tibia of each Kymouse were processed into single-cell suspension as described above. From these 22 bone marrow samples, seven were FACS sorted to recover CD19$^+$ B220$^+$ B cells into a single tube. The cells were lysed and RT-PCR was performed to amplify the V$_H$ domain, followed by standard 2×300 bp Illumina library generation, before sequencing on an Illumina MiSeq. The pRESTO and Change-O pipelines were used to process the sequences generated by the MiSeq sequencers (*Vander Heiden et al., 2014*; *Gupta et al., 2015*). Change-O's MaskPrimers was used to assign isotypes; germline genes were assigned and identity to germline was calculated using Change-O's AssignGenes with the Kymouse germline database. All IgM sequences with zero mutations were selected for further analysis resulting in a total of 412,493 V$_H$ sequences across the seven Kymice (average: 58,928; range: 31,905–100,240).

## NGS sequence analysis of V$_H$ sequences derived from human samples

Buffy coat samples were obtained from 10 healthy individuals as described previously (*Ghraichy et al., 2021*). In the previous study, B cells were FACS sorted into naïve, marginal zone, plasma, and switched memory cell populations. In the present study, we re-analysed a subset of these sequences, which were IgM sequences from the naïve subset of B cells with zero mutations. Isotypes were used as assigned in the previous study. Germline genes were assigned and germline identity were calculated using Change-O's AssignGenes with the human reference IMGT database (*Gupta et al., 2015*; *Giudicelli et al., 1997*). There was a total of 338,677 sequences (mean: 33,867 per human, range: 20,653–48,293).

## NGS sequence analysis of VH sequences derived from C57BL/6 WT mice

For a comparator for CDRH3 sequence and structural sharing, we used a previously published dataset of high-depth bulk V$_H$ sequencing from C57BL/6 mice (*Greiff et al., 2017*). This is a reasonable comparator as the Kymouse is bred from a mixture of C57BL/6 and 129S7 strains. The ideal comparator would be 129S7 as the Kymouse constant region genes are derived from the 129S7 strain, however we could not find equivalent publicly available data for this strain. Furthermore, the C57BL/6 strain is best characterised immunogenetically and at the BCR repertoire level (*Jackson et al., 2022*; *Greiff et al., 2017*).

We downloaded 6,763,480 IgM V$_H$ nucleotide sequences from naïve B cells of healthy unvaccinated C57BL/6 wild-type mice from the Observed Antibody Space (OAS) (*Greiff et al., 2017*; *Kovaltsuk et al., 2018*; *Olsen et al., 2022*). Those sequences with any nucleotide mutations were removed, and the remaining sequences were down-sampled via stratified sampling aiming to preserve the original clonal structure in the complete dataset; 150,000 sequences with redundancy were randomly selected from each of the five C57BL/6 mice. Collapsing to unique (nucleotide) sequences resulted in a total of 268,285 sequences (mean: 53,657 sequences per mouse; range: 20,026–87,041).

## Clonal and diversity analysis of NGS sequence data

Clonotypes are defined as sequences with common IGHV and IGHJ genes and 90% or more amino acid identity across length-matched CDRH3s. Antibody sequences were assigned to clonotypes using the DefineClones module of Change-O (*Gupta et al., 2015*) under the amino acid model.

Shannon diversity (H) was calculated using the *stats.entropy* function within Python's SciPy library. The formula is as follows:

$$H = -\sum_{i=1}^{s} p_i ln p_i$$

where p$_i$ is the proportion of sequences in the clonotype *i* of *s* clonotypes.

## Structural annotation

Productive sequences were translated into amino acids and IMGT numbered (*Lefranc et al., 2003*) using ANARCI (*Dunbar and Deane, 2016*). IMGT CDR sequences were extracted and structurally annotated using the SAAB+ version 1.01 pipeline (*Kovaltsuk et al., 2020*). SAAB+ uses SCALOP (*Wong et al., 2019*) to assign each sequence's CDRH1 and CDRH2 loops to a structural canonical class and uses FREAD (*Choi and Deane, 2010*) to identify if the CDRH3 loop has a similar structure to any crystallographically solved set of 4544 CDRH3 structures (referred to as templates, downloaded from SAbDab [*Dunbar et al., 2014*; *Schneider et al., 2022*] on 16 February 2022).

To reduce dimensionality, templates are clustered with a 0.6 Å RMSD cut-off, producing a set of 1944 templates. In this set of templates, 41% of the antibody structures are of murine-origin and 37% are of human-origin (*Figure 6—figure supplement 3*). The SAAB+ pipeline outputs for each sequence the canonical class of the CDRH1 and CDRH2 loops, and the Protein Data Bank ID of the structure that contains the best matched CDRH3 structure for homology modelling from the 1944 templates.

As a complementary approach, we also modelled representatives of all non-singleton clonotypes in each of the repertoires using a recent deep learning method, ABlooper (*Abanades et al., 2022*). For full structural modelling with ABlooper, a light chain must be supplied. As we did not know the cognate light chain for any of the heavy chains from bulk sequencing, all heavy chains were paired with a single light chain which was chosen to be the most commonly observed light chain in the Kymouse dataset. We selected a common light chain to standardise its effect on the prediction of the heavy chain CDRs, in the absence of knowledge of the true light chain.

## Humanness scoring of $V_H$ sequences

The human, Kymouse and C57BL/6 mouse $V_H$ sequences were assigned a 'humanness score' using the RF regressors from Hu-mAb (*Marks et al., 2021*). Sequences were first IMGT numbered using ANARCI as above. While the C57BL/6 mouse and Kymouse sequences were mostly full length (IMGT positions 1–128), the human sequences were in most cases missing FWR1 (IMGT positions 1–26 under the IMGT CDR definition). As the human sequences were non-mutated, we considered it reasonable to simply fill in FWR1 according to the sequence found in the assigned germline. For the human and Kymouse sequences, the RF model trained on the IGHV gene assigned by IgBLAST were used for scoring, while for the murine sequences all seven (IGHV1-7) RF models were used to score the sequence and the highest score was selected. We used the IGHV-specific classification thresholds reported in the Hu-mAb paper to annotate if a sequence was considered human or not (*Marks et al., 2021*).

## Immunophenotyping of Kymice

Spleens, lymph nodes, and bone marrow from a further 12 antigen-inexperienced Kymice were processed to single-cell suspensions and cryopreserved as described above. FACS of the bone marrow samples was performed using fluorescently conjugated antibodies against B220, IgM, IgD, IgG1, IgG2ab, IgG3, CD8, CD4, Ly-6G, CD11c, and CD138 (BD Biosciences), CD19, F4/80, Sca-1 (BioLegend), and TACI (eBioscience). For the pooled spleen and lymph node samples the FACS panel consisted of B220, IgM, IgD, IgG1, IgG2ab, IgG3, CD8, CD4, Ly-6G&6C, CD11c, and CD95 (BD Biosciences), CD19, F4/80, CD73, CD80, PD-L2, and GL7 (BioLegend). DRAQ7 (BioStatus) was used in all samples to distinguish live and dead cells. For flow cytometry, cells were thawed from frozen, resuspended in warm 10% FBS in RPMI buffer, filtered through 40 µm cell strainers, and centrifuged at $400 \times g$ for 10 min at 4°C. Cells were resuspended in buffer and TrueStain FcX (BioLegend) was added for 10 min on ice. Single-cell suspensions of bone marrow cells and pooled spleen and lymph node cells were stained with their respective staining cocktails for 30 min. All cells were spin washed and resuspended in buffer, filtered through a 30 µm cell strainer, followed by cell sorting on a 5-laser BD FACS Aria Fusion flow cytometer (Beckton Dickinson).

## Analysis of flow cytometry data from Kymice

The frequencies of the following cell types within total viable (DRAQ7) cells were determined using classical FACS gating: bone marrow plasma cells (CD138+TACI+/Sca-1+), spleen/lymph node memory B cells (B220+CD19+IgD-CD73+/CD80+/PDL2+), and spleen/lymph node GC cells (B220+CD19+CD95+GL7+; data not shown). Cell populations were also analysed in an unbiased manner using

unsupervised clustering algorithms. In brief, the raw.fcs files were imported into R (RStudio version 1.2.5033 with R version 4.0.0) using CytoExploreR (version 1.0.8) and the data were transformed to normalise marker intensities (logicle transform). For visualisation, additional quantile scaling from 0 to 1 was performed, fixing values less than the 1st percentile to 0.01 and values above the 99th percentile to 0.99 to minimise the contribution of outliers to the scaling. Cell clustering was performed using the Leiden clustering algorithm (R package Monocle 2.16) and clusters were visualised in two-dimensional space using UMAP (R package uwot 0.1.10). Poorly resolved clusters were re-clustered, the subclusters manually merged to the first level clusters and annotated by cell type.

## Code availability

The Python code used to analyse the data and generate the figures is available at https://github.com/oxpig/HumMus (copy archived at *Richardson, 2023*).

## Acknowledgements

This work was funded by the Bill & Melinda Gates Foundation, OPP1159947. The funder did not play any role in the study design, data collection and analysis, decision to publish, or preparation of the manuscript. ER is funded by the Medical Research Council (grant number: MR/R015708/1).

## Additional information

### Competing interests

Eve Richardson: Eve Richardson receives funding from Kymab Ltd. Špela Binter: Špela Binter is an employee of Kymab Ltd. Miha Kosmac: Miha Kosmac was an employee of Kymab within the last three years. Jacob D Galson: Employee of Alchemab Therapeutics Ltd. Paul Kellam: Paul Kellam is an employee of Kymab Ltd. Simon J Watson: Simon Watson is an employee of Kymab Ltd. The other authors declare that no competing interests exist.

### Funding

| Funder | Grant reference number | Author |
| --- | --- | --- |
| Bill & Melinda Gates Foundation | OPP1159947 | Paul Kellam |
| Medical Research Council | MR/R015708/1 | Eve Richardson |

The funders had no role in study design, data collection and interpretation, or the decision to submit the work for publication.

### Author contributions

Eve Richardson, Conceptualization, Data curation, Software, Formal analysis, Visualization, Writing – original draft, Writing – review and editing; Špela Binter, Conceptualization, Data curation, Formal analysis, Visualization, Writing – original draft, Writing – review and editing; Miha Kosmac, Data curation, Formal analysis, Investigation, Writing – review and editing; Marie Ghraichy, Valentin von Niederhäusern, Johannes Trück, Data curation, Writing – review and editing; Aleksandr Kovaltsuk, Jacob D Galson, Conceptualization, Data curation, Writing – review and editing; Dominic F Kelly, Charlotte M Deane, Paul Kellam, Supervision, Writing – review and editing; Simon J Watson, Conceptualization, Data curation, Formal analysis, Writing – original draft, Writing – review and editing

### Author ORCIDs

Eve Richardson ⓘ http://orcid.org/0000-0002-5499-6283
Miha Kosmac ⓘ http://orcid.org/0000-0002-5372-7233
Valentin von Niederhäusern ⓘ http://orcid.org/0000-0001-9150-2361
Aleksandr Kovaltsuk ⓘ http://orcid.org/0000-0003-3806-8302
Jacob D Galson ⓘ http://orcid.org/0000-0003-4916-800X
Johannes Trück ⓘ http://orcid.org/0000-0002-0418-7381
Charlotte M Deane ⓘ http://orcid.org/0000-0003-1388-2252

## Ethics

This study was carried out under Project Licenses 70/8718 issued by the UK Government Home Office under Animal (Scientific Procedures) Act (A(SP)A), 1986, incorporating Directive 2010/63/EU of the European Parliament, and with the approval of the Sanger Institute Animal Welfare and Ethical Review Body. The Institute complied with the Code of Practice issued by the UK Government which aids compliance with the A(SP)A. The Institute has a PHS assurance F16-00128 (WTSI).

## Decision letter and Author response

Decision letter https://doi.org/10.7554/eLife.81629.sa1
Author response https://doi.org/10.7554/eLife.81629.sa2

# Additional files

## Supplementary files

• Transparent reporting form

• Supplementary file 1. Supplementary Tables. Supplementary File 1a: Dominant canonical forms per IGHV subgroup. Predicting the canonical form based solely on the dominant form observed for the IGHV subgroup would result in accuracy between 46.2% and 100%. Differences can be seen between the Kymouse and human in IGHV1, in which the majority of human sequences are H1-8-A vs. H1-8-C in the Kymouse. Supplementary File 1b: Information about the number of CDRH3 structural clusters produced with different thresholds under the select greedy clustering algorithm. 0.6 Å was the threshold selected by *Kovaltsuk et al., 2020*, in the original SAAB+ application.

## Data availability

The processed paired sequencing data is available at https://doi.org/10.5281/zenodo.7474233 and the processed bulk VH sequence data has been deposited in the Observed Antibody Space (http://opig.stats.ox.ac.uk/webapps/oas/oas). Immunophenotyping data is available in the FlowRepository (https://flowrepository.org/) under IDs FR-FCM-Z5LB (bone marrow samples) and FR-FCM-Z5LC (splenocyte and lymph node samples).

The following datasets were generated:

| Author(s) | Year | Dataset title | Dataset URL | Database and Identifier |
|---|---|---|---|---|
| Richardson E, Binter S, Watson SJ | 2022 | IGM sequences from naïve Kymouse 1 | http://opig.stats.ox.ac.uk/webapps/oas/dataunit?unit=Richardson/csv/Mouse-1_Richardson_2022_1_Heavy_IGHM.csv.gz | Observed Antibody Space, Mouse-1_Richardson_2022_1_Heavy |
| Binter S, Watson SJ | 2022 | Characterisation of the immune repertoire of a humanised transgenic mouse through immunophenotyping and high-throughput sequencing (bone marrow) | http://flowrepository.org/id/FR-FCM-Z5LB | FlowRepository, FR-FCM-Z5LB |
| Binter S, Watson SJ | 2022 | Characterisation of the immune repertoire of a humanised transgenic mouse through immunophenotyping and high-throughput sequencing (spleen) | http://flowrepository.org/id/FR-FCM-Z5LC | FlowRepository, FR-FCM-Z5LC |

*Continued on next page*

*Continued*

| Author(s) | Year | Dataset title | Dataset URL | Database and Identifier |
|---|---|---|---|---|
| Richardson E, Binter S, Watson SJ | 2022 | IGM sequences from naïve Kymouse 2 | http://opig.stats.ox.ac.uk/webapps/oas/dataunit?unit=Richardson/csv/Mouse-2_Richardson_2022_1_Heavy_IGHM.csv.gz | Observed Antibody Space, Mouse-2_Richardson_2022_1_Heavy |
| Richardson E, Binter S, Watson SJ | 2022 | IGM sequences from naïve Kymouse 3 | http://opig.stats.ox.ac.uk/webapps/oas/dataunit?unit=Richardson/csv/Mouse-3_Richardson_2022_1_Heavy_IGHM.csv.gz | Observed Antibody Space, Mouse-3_Richardson_2022_1_Heavy |
| Richardson E, Binter S, Watson SJ | 2022 | IGM sequences from naïve Kymouse 4 | http://opig.stats.ox.ac.uk/webapps/oas/dataunit?unit=Richardson/csv/Mouse-4_Richardson_2022_1_Heavy_IGHM.csv.gz | Observed Antibody Space, Mouse-4_Richardson_2022_1_Heavy |
| Richardson E, Binter S, Watson SJ | 2022 | IGM sequences from naïve Kymouse 5 | http://opig.stats.ox.ac.uk/webapps/oas/dataunit?unit=Richardson/csv/Mouse-5_Richardson_2022_1_Heavy_IGHM.csv.gz | Observed Antibody Space, Mouse-5_Richardson_2022_1_Heavy |
| Richardson E, Binter S, Watson SJ | 2022 | IGM sequences from naïve Kymouse 6 | http://opig.stats.ox.ac.uk/webapps/oas/dataunit?unit=Richardson/csv/Mouse-6_Richardson_2022_1_Heavy_IGHM.csv.gz | Observed Antibody Space, Mouse-6_Richardson_2022_1_Heavy |
| Richardson E, Binter S, Watson SJ | 2022 | IGM sequences from naïve Kymouse 7 | http://opig.stats.ox.ac.uk/webapps/oas/dataunit?unit=Richardson/csv/Mouse-7_Richardson_2022_1_Heavy_IGHM.csv.gz | Observed Antibody Space, Mouse-7_Richardson_2022_1_Heavy |
| Richardson E, Binter S, Watson SJ | 2022 | Single-cell naïve IgM VH:VL sequence data from 22 Kymice | https://zenodo.org/record/7474233#.Y6S1NuzP3DE | Zenodo, 10.5281/zenodo.7474233 |

The following previously published datasets were used:

| Author(s) | Year | Dataset title | Dataset URL | Database and Identifier |
|---|---|---|---|---|
| Ghraichy M, Trück J | 2021 | Pre-processed IgH repertoire sequencing data from BioProject PRJNA748239 | https://zenodo.org/record/5155565#.YuFlHy2ZPdc | Zenodo, 10.5281/zenodo.5155565 |
| Greiff V, Menzel U, Miho E, Weber C, Riedel R, Cook S, Valai A, Lopes T, Radbruch A, Winkler TH, Reddy ST | 2017 | Antibody repertoire RNA-seq throughout key stages of mouse B-cell development (pre-B-cells, naive B cells, plasma cells) | https://www.ebi.ac.uk/arrayexpress/experiments/E-MTAB-5349/ | ArrayExpress, E-MTAB-5349 |

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
