## [Editor Report]

This is an important study that defines the limits of using human Ig sequences in genetically modified mice to explore human immune responses. The study was carefully designed, and the results should be of interest to a wide readership and will be important for grounding future studies/uses of the Ky mouse.

---

## [Decision Letter]

**Decision letter after peer review:**

Thank you for submitting your article "Characterisation of the immune repertoire of a humanised transgenic mouse through immunophenotyping and high-throughput sequencing" for consideration by *eLife*. Your article has been reviewed by 3 peer reviewers, including Ranjan Sen as the Reviewing Editor and Reviewer #1, and the evaluation has been overseen by a Reviewing Editor and Satyajit Rath as the Senior Editor.

Essential revisions:

1) Please address all of each reviewer's comments for style and clarity (noted under 'Recommendations for Authors' in the reviews below).

2) Please provide additional analysis of Ig light chain (IGK and IGL) repertoires in Ky mice to complement the IgH analyses in the current manuscript. Is there a strong rationale against providing a complete analysis (IgH^+^IgL) and assessment of this model?

3) Provide a more thorough discussion of the potential uses and limitations of this animal model in evaluating human immune responses.

*Reviewer #1 (Recommendations for the authors):*

Additional clarifications of previous studies of immune responses in Ky mice compared to humans.

*Reviewer #2 (Recommendations for the authors):*

– Experimental setup figure needed.

– Figure 1: First sentence should be a result with a verb. Unclear from the caption why this is figure 1. Please make this more clear (same for all other figures).

– Figure 2: can you show these plots also as bar plots (in the supplementary) and provide significance analysis? Relatedly, the entire section of the main text on V gene usage does not contain one number. You need to quantify…

– Figure 3B: I don't understand the y-axis numbers…please clarify in the text.

– Figure 4: what's the clonal overlap across mice? In Greiff et al. Cell Reports 2017, the overlap was quite high…it should be lower here since human repertoires are more diverse… And what's the overlap on the structural level?

– In this work, you did not discuss whether antigen-specific responses are similar in hu-mice and humans. Can you comment on this?

*Reviewer #3 (Recommendations for the authors):*

1. Lines 35-36: This statement should be reworded; authors should be precise. This mouse does not have "the full set of variable genes", it's in fact missing many known IGHV genes. More accurate to say that it was the first mouse in which an entire IGH haplotype was integrated or something along those lines.

2. Line 54: The term "BCRs" has already been defined on line 52.

3. Lines 81-82: "V", "D", and "J" already defined above.

4. Line 123: How were spleen and lymph node populations pooled?

5. Lines 132-136 and 141-145: More detail is needed here. What protocol was used to conduct RT-PCR and generate VDJ libraries? What type of MiSeq libraries? 2x300 bp? Where UMIs used? I presume that PRESTO was also used? This should be stated as well as other parameters used/details regarding sequence read processing.

6. Lines 140 and 149: Should be "FACS"?

7. Lines 144: A description of processing steps leading to the 412,493 sequences would be useful here. (Same comment for human data, line 151).

8. Section starting at Line 147: Can you be more explicit? Did you generate additional data from the same samples used by Ghraichy et al? Or did you reanalyze data from this study?

9. Section starting at line 154: Is C57BL/6 the most appropriate strain for comparison? Are the constant gene regions from 129S7? Some rationale/discussion should be provided here regarding the genetic background of the Kymouse.

10. Line 164: "IGHV" and "IGHJ" are not defined.

11. Line 199: Sometimes "VH" is used, sometimes IGHV, etc. Maybe either choose one convention or be clear about why one or the other is used.

12. Line 204: What was done in cases for which the germline assignment was ambiguous due to variation not captured in FWR1?

13. Lines 243-244: It's not clear what data is available on opig? Authors should outline what is provided. It appears to be processed data. Is the raw data available on any other public repository? E.g., EMBL, NCBI?

14. Line 262: You reference Figure 1C before 1B in the text. The panel order should probably be switched. In addition, it looks like the figures in panels 1C and 1D are currently in reverse order. The labels in panel D seem to match panel A, and C appears to match panel B.

15. Lines 318-319: I believe the authors mean "dendrogram" rather than "phylogenetic tree".

16. Lines 319-320: It is not necessarily the case that the ky mouse and human are the "same" in the number of genes used, which is quite clear from the heatmap in panel 2A. The authors should be more precise in their language. As noted earlier, the Kymouse does not carry a full set of all human IGHV genes but instead carries IGHV genes found on just the human haplotype that was used to construct the mouse. Your data clearly show this, e.g., based on your "outlier" sample, which is probably the only sample in your cohort that carries IGHV4-38-2. Based on the analysis by Lee et al. 2014, it would appear that the Kymouse does not carry IGHV4-38-2. The authors should not overgeneralize.

17. Is there a reason the authors did not also compare gene usage profiles for IGK and IGL datasets to control human samples?

18. Lines 343-350: It would seem appropriate to also compare CDR3 lengths at an individual level, rather than pooling data across all individuals in each species and taking the average. Again, the assumption is being made that all humans come from the same distribution – pooling the data across all human samples, for example, makes it difficult to assess inter-individual variation. It would be worthwhile at least provide plots and summary data for this in the supplement.

19. Line 540: "BCR" already defined previously.

20. Lines 557-559: It could also be worth noting that these transgenic animals may in fact poorly represent the full extent of responses (in terms of diversity) observed across the human population. This will undoubtedly deserve attention as data from these and other transgenic animal models are interpreted in the future. It would be interesting to understand the authors' viewpoints on what uses or limitations the Kymouse (and other similar transgenic animals) might have.

---

## [Author Response]

Essential revisions:1) Please provide additional analysis of Ig light chain (IGK and IGL) repertoires in Ky mice to complement the IgH analyses in the current manuscript. Is there a strong rationale against providing a complete analysis (IgH^+^IgL) and assessment of this model?

We thank the reviewers for highlighting that this missing analysis is important for full analysis of the Kymouse BCR repertoires.

To this end, we have performed germline gene usage analysis on the cognate light chains in the paired Kymouse data which we did not make full use of, as pointed out in another comment (we previously only used this single-cell data to calculate the kappa:λ ratio).

We analysed an extra 2 paired repertoires in addition to this data, and examined it together with data from a recent excellent high-depth paired sequencing study by Jaffe and colleagues, from which were able to select data from cells that had been sorted in a similar way for naïve status (doi: 10.1038/s41586-022-05371-z).

As the bulk VH sequencing correlated well with the paired sequencing in terms of subgroup gene frequencies, we believe that the more complete analysis of our paired dataset provides a sufficient description of the light chain repertoires of the Kymice.

The novel analyses and corresponding Methods are presented here.

Methods

“Next Generation Sequencing (NGS) Analysis of Paired V_H_ and V_L_ Sequences from Single-Cell Sorted B Cells Derived from Kymice

For 22 Kymice, the spleen and inguinal lymph nodes were processed to single-cell suspensions as described above before fluorescently-activated cell sorting (FACS) to recover CD19^+^ B220^+^ B cells into individual wells of a 96 well plate. RT-PCR was performed to amplify the V_H_ and V_L_ domains, and standard 2x300bp Illumina libraries were generated before sequencing on an Illumina MiSeq sequencer as previously described (Lee et al. 2014). Unique molecular identifiers (UMIs) were used to group derivatives of the same RNA molecules.

pRESTO was used to process the reads and build UMI consensus groups (Vander Heiden et al. 2014). The Change-O pipeline (Gupta et al. 2015) was then used to process the resultant sequences; naïve BCR sequences were characterised as IgM sequences containing zero nucleotide mutations. In total, *3,*175 paired V_H_ and V_L_ sequences were processed from the 22 Kymice. In subsequent analyses comparing gene frequencies, we removed data from seven Kymice where fewer than 100 paired sequences were retrieved, resulting in 15 Kymice contributing to the final dataset. In these *15* Kymice, the average sample size was 202 (range: 105 to 363).”

NGS Sequence Analysis of Published Paired V_H_ and V_L_ Sequences from Naïve Human B cells

As a human comparator for the single-cell Kymouse data, we used a recently published, high-depth paired sequencing dataset, selecting the subset of IgM sequences with zero mutations from naïve B-cells as made available by Jaffe and colleagues (Jaffe et al. 2022). We removed sequences from cells in which there were multiple heavy or light chains. This resulted in 478,177 unique total sequences across four subjects (average: 119,544 sequences per subject; range: 88,631 – 143,293).

NGS Sequence Analysis of V_H_ Sequences from Bulk Sorted B Cells Derived from Kymice

Bone marrows from the femur and tibia of each Kymouse were processed into single-cell suspension as described above. From these 22 bone marrow samples, seven were FACS sorted to recover CD19^+^ B220^+^ B cells into a single tube. The cells were lysed and RT-PCR was performed to amplify the V_H_ domain, followed by standard 2x300bp Illumina library generation, before sequencing on an Illumina MiSeq. The pRESTO and Change-O pipelines were used to process the sequences generated by the MiSeq sequencers (Vander Heiden et al. 2014; Gupta et al. 2015). IgM sequences with zero mutations were selected for further analysis resulting in a total of 412,493 V_H_ sequences across the seven Kymice (average: 58,928; range: 31,905 –100,240).”

Results

“The Kymouse naïve antibody sequence repertoire is more human-like than murine-like

Using high-throughput paired sequencing we recovered 3,175 full-length paired IgM V_H_ and V_L_ sequences and a further 451,655 full-length unpaired IgM V_H_ sequences from naïve B cells extracted from the spleens and lymph nodes of 22 Kymice.

In order to evaluate the humanness of the Kymouse naïve B cell sequence repertoire, we performed two comparisons: a comparison of the lower-depth single-cell paired sequencing data with a published, high-depth paired human naïve dataset, and a more extensive comparison of the bulk V_H_ sequences to equivalent datasets of 338,677 V_H_ sequences from human naive B cells, and 268,285 V_H_ sequences from C57BL/6 mice.

[…]

We further examined the distributions of CDRH3 and CDRL3 length (Figure 2F). We found that human CDRH3s were on average longer than Kymouse CDRH3s (16.4aa ± 0.01 vs. 14.5aa ± 0.13); there was no significant difference in CDRL3 length (average 9.8aa ± 0.002 vs 9.7aa ± 0.2). To identify possible causes for a difference in CDRH3 length, we compared IGHD gene alignment length and length of nucleotide insertions at the VD and DJ junctions. While there was a small but significant (5% level) difference in IGHD gene alignment length (average 4.6aa ± 0.06 vs. 4.4aa ± 0.08), the largest difference is in insertion length at both the VD and DJ junctions (average 7.8 nt ± 0.01 vs. 4.1nt ± 0.1, 7.3nt ± 0.02 vs. 3.7nt ± 0.1). A significantly greater proportion of the Kymouse primary repertoire does not have junctional insertions compared to the human repertoire: 19.1% of the Kymouse sequences had no VD insertions, while 8.2% of the human sequences lacked VD insertions. 14.1% of Kymouse sequences had no DJ insertions while 4.5% of human sequences lacked these insertions. Over 10 times as many of the sequences in the Kymouse repertoires lacked junctional insertions compared to human repertoires (3.8% in Kymice vs. 0.3% in humans).”

2) Provide a more thorough discussion of the potential uses and limitations of this animal model in evaluating human immune responses.

We agree with the reviewer that this is an important take-away from the paper. As a result, we have added a section in the Discussion and expanded on another Discussion section.

Line 1469:

“Kymice contain the immunoglobulin haplotype of a single human subject consistent with reduced diversity relative to outbred human B cell repertoires where the considerable allelic and haplotypic diversity observed in human populations has significant effects on the expressed BCR repertoire (Rodriguez et al. 2022). Given the Kymouse’s use of a single haplotype, and differences in the observed germline gene usages, it is likely that the BCR response to antigen stimulus in the Kymouse will represent only a subset of the possible diversity of the outbred human response and this should be consideration in vaccine response modelling.”

Reviewer #1 (Recommendations for the authors):Additional clarifications of previous studies of immune responses in Ky mice compared to humans.

We have added further clarification via expanding on the section in the Discussion section, including all relevant literature.

Line 1476:

“In conclusion, although naïve BCR repertoires of the Kymouse have key distinctions from human repertoires at the sequence level they are comparable to the human repertoires in terms of CDRH3 structural usage. The less diverse primary repertoire of the Kymouse does not appear to be an impediment to its use as a source of high-affinity therapeutic antibodies, and indeed a higher frequency of shorter CDRH3 lengths could be favorable in terms of the developability profile of any mAbs derived from these systems (Raybould et al. 2019). In addition to the analysis of naïve, antigen-inexperienced repertoires in the current study, we and others have also used the Kymouse as a model system in multiple immunisations with a broad range of different viral, bacterial and parasite antigen targets, as well as self-antigens for therapeutic applications (Walls et al. 2020; Jiao et al. 2020; Sainson et al. 2020; Petzer et al. 2020; Hastie et al. 2021; Richardson et al. 2021; Saghari et al. 2022). The polyclonal serum response elicited in these Kymice has been shown to be functional and the isolated monoclonal antibodies bind physiologically important epitopes recognised by similar human antibodies (Rujas et al. 2020). Indeed, a number of studies have showed the Kymouse is able to elicit monoclonal antibodies equivalent to those found in humans exposed to the same antigen (Sok et al. 2016; Scally et al. 2017; McLeod et al. 2019; Oyen et al. 2020). Together these data suggest that the engagement of Kymouse BCRs on naïve B cells is authentic, and that the structural templates available for antigen binding are indeed human-like.”

Reviewer #2 (Recommendations for the authors):– Experimental setup figure needed.

We apologise for the lack of an experimental setup figure in the original manuscript. We have added Figure 7:

– Figure 1: First sentence should be a result with a verb. Unclear from the caption why this is figure 1. Please make this more clear (same for all other figures).

We thank the reviewer for pointing this out and have amended all figure captions such that the first sentence describes a result.

– Figure 2: can you show these plots also as bar plots (in the supplementary) and provide significance analysis? Relatedly, the entire section of the main text on V gene usage does not contain one number. You need to quantify…

We apologise for the overly qualitative nature of this section of the text and have addressed this comment by producing the relevant bar plots, significance analysis and adding more figures into the text. Statistical analysis was in the form of Bonferroni-corrected t-tests on gene frequencies at the level of gene subgroups.

Main text:

“We used the bulk V_H_ data to re-calculate gene usage frequencies in Kymice at a higher depth, and compare the usage frequency of the IGHV, IGHD, and IGHJ germline genes to those in bulk heavy chain human data of comparable depth. At such enhanced depth, all IGHD and IGHJ genes in the Kymouse haplotype were observed in all repertoires, as were all non-orphon IGHD and IGHJ genes in the general human database. There were thirteen IGHV genes found in human repertoires that are absent in the Kymouse haplotype which were observed in some but not all human repertoires (Figure 3—figure supplement 1).

Statistical testing on a gene-by-gene basis was largely in agreement with the findings from the single-cell analysis (Figure 3—figure supplement 2) with the exception that IGHJ4 usage was elevated in the bulk data relative to the single-cell data, and IGHJ6 usage lower such that it is not significantly different from human repertoires (Figure 3—figure supplement 3).

In addition to statistical testing on a gene-by-gene basis (Figure 3—figure supplement 2), we used hierarchical clustering to compare the gene usage profiles of individual Kymice and humans, building dendrograms to show the relationships between the individuals’ gene usage profiles. The frequencies as determined by sequence abundance are shown to differ in such a way that the two repertoire types can be clustered separately. The hierarchical clustering of the IGHV genes showed that the Kymice and humans form nearly separate monophyletic clusters except for a single outlier human subject (Figure 3A). Most of the variation in IGHV gene usage is explained by the IGHV gene subgroup usage: clustering by IGHV gene subgroup usage separates humans and Kymice without the outlier human sample, with Kymice using a lower proportion of IGHV1 and IGHV2 genes (5.3% vs. 22.6% and 0.2% vs. 2.1% in Kymouse vs. human repertoires respectively for these subgroups), and an increased IGHV3 (51.6% vs. 39.7%), IGHV4 (35.3% vs. 31.6%) and IGHV6 usage (4.7% vs. 1.0%) (Figure 3B).

The IGHJ gene usage profile is similar: Kymice and nine out of the ten humans form monophyletic clades with a single outlier human. On average, the Kymouse uses IGHJ4 more frequently than humans (46.7% vs. 42.6%), and IGHJ5 and IGHJ1 less frequently (10.1% vs. 13.5%, and 0.6% vs. 1.0%) (Figure 3C and Figure 3—figure supplement 3). Both the IGHV and IGHJ gene usage profiles of naïve Kymouse repertoires are more similar to one another than to any human repertoire.

The IGHD gene usage is likewise distinguishable between humans and Kymice on average (Figure 3D). As can be seen from the heatmap, the IGHD germline genes used by the Kymice (*e.g.* IGHD3-22, IGHD2-15) are infrequently used by humans and *vice versa*. IGHD2, IGHD4 and IGHD5 subgroups are preferred by human repertoires (average 17.2% vs. 5.8%, 9.1% vs. 5.5% and 9.5% vs. 4.8% respectively in human vs. Kymouse repertoires), while Kymouse repertoires preferentially use IGHD1, IGHD6 and IGHD7 (24.2% vs. 10.3%, 27.4% vs. 20.4% and 2.4% vs. 0.6% on average in Kymice and humans respectively) (Figure 3—figure supplement 2).”

– Figure 3B: I don't understand the y-axis numbers…please clarify in the text.

contribution of each of the five contributing genetic factors to the CDRH3 length difference observed between Kymice and humans: where positive, it means that the factor results in longer CDRH3s on average in the Kymouse repertoire vs. the human repertoire (i.e. V gene choice in this case), and where negative, it contributes to the described shorter CDRH3s in the Kymice on average.

We have changed the figure legend to clarify.

The y-axis in Figure 3B shows the estimated contribution of each of the five contributing genetic factors to the CDRH3 length difference observed between Kymice and humans: where positive, it means that the factor results in longer CDRH3s on average in the Kymouse repertoire vs. the human repertoire (i.e. V gene choice in this case), and where negative, it contributes to the described shorter CDRH3s in the Kymice on average.

We have changed the figure legend to clarify:

– Figure 4: what's the clonal overlap across mice? In Greiff et al. Cell Reports 2017, the overlap was quite high…it should be lower here since human repertoires are more diverse… And what's the overlap on the structural level?

We agree with the reviewer that this is an important piece of missing information in this section and thank them for this suggestion. With regards to the clonal overlap across mice, we found that is significantly greater than among humans, but less than among Kymice. With regards to the overlap on the structural level, overlap on a structural level in terms of the % shared CDRH3 structural clusters is much higher between any two subjects than clonotype sharing, as is to be expected given the reduced dimensionality. The structural sharing is highest between individual Kymice and individual mice, as predicted from sharing of CDRH3 sequences and clonotypes; it is lowest between Kymice and mice, which is not necessarily reflected in the CDRH3 sequence sharing. The structural sharing between the humans and Kymice is comparable to from human to human. This adds an extra detail to the structural comparison.

Line 1088

“Interestingly, clonotype sharing was significantly higher among individual Kymice (average: 7.92%, range: 6.04 – 11.14%) than between individual mice (average 4.55%, range: 3.52% – 5.90%) despite comparable CDRH3 sharing.”

The overlap on a structural level in terms of the % shared CDRH3 structural clusters is much higher between any two subjects than clonotype sharing, as is to be expected given the reduced dimensionality. As an interesting comparison to this pairwise CDRH3 and clonotype sharing, we provide two plots in Figure 6—figure supplement 5: one showing sharing of CDRH3 structural clusters (analogous to CDRH3 sequence sharing) and one showing sharing of “structural clonotypes”, which is a combination of H1 and H2 canonical classes and the H3 structural cluster (analogous to sequence clonotypes).

– In this work, you did not discuss whether antigen-specific responses are similar in hu-mice and humans. Can you comment on this?

We agree that this is an important topic that we did not discuss, and have added the following to the Discussion section in order to address this.

Lines 1476

“In conclusion, although naïve BCR repertoires of the Kymouse have key distinctions from human repertoires at the sequence level they are comparable to the human repertoires in terms of CDRH3 structural usage. The less diverse primary repertoire of the Kymouse does not appear to be an impediment to its use as a source of high-affinity therapeutic antibodies, and indeed a higher frequency of shorter CDRH3 lengths could be favorable in terms of the developability profile of any mAbs derived from these systems (Raybould et al. 2019). In addition to the analysis of naïve, antigen-inexperienced repertoires in the current study, we and others have also used the Kymouse as a model system in multiple immunisations with a broad range of different viral, bacterial and parasite antigen targets, as well as self-antigens for therapeutic applications (Walls et al. 2020; Jiao et al. 2020; Sainson et al. 2020; Petzer et al. 2020; Hastie et al. 2021; Richardson et al. 2021; Saghari et al. 2022). The polyclonal serum response elicited in these Kymice has been shown to be functional and the isolated monoclonal antibodies bind physiologically important epitopes recognised by similar human antibodies (Rujas et al. 2020). Indeed, a number of studies have showed the Kymouse is able to elicit monoclonal antibodies equivalent to those found in humans exposed to the same antigen (Sok et al. 2016; Scally et al. 2017; McLeod et al. 2019; Oyen et al. 2020). Together these data suggest that the engagement of Kymouse BCRs on naïve B cells is authentic, and that the structural templates available for antigen binding are indeed human-like.”

Reviewer #3 (Recommendations for the authors):1. Lines 35-36: This statement should be reworded; authors should be precise. This mouse does not have "the full set of variable genes", it's in fact missing many known IGHV genes. More accurate to say that it was the first mouse in which an entire IGH haplotype was integrated or something along those lines.

We thank the reviewer for highlighting this inaccuracy and for their suggested more precise alternative wording. We have changed the text in the relevant section.

Line 39

“Twenty-five years of progress in genetic engineering since the first Ig transgenic mouse (M. Brüggemann et al. 1989) culminated in 2014 in the integration of a complete human immunoglobulin (Ig) haplotype in mice for the first time (Lee et al. 2014).”

2. Lines 343-350: It would seem appropriate to also compare CDR3 lengths at an individual level, rather than pooling data across all individuals in each species and taking the average. Again, the assumption is being made that all humans come from the same distribution – pooling the data across all human samples, for example, makes it difficult to assess inter-individual variation. It would be worthwhile at least provide plots and summary data for this in the supplement.

We agree that this is an interesting plot and pooling the data could mask significant interindividual variation. Conceivably, individuals could differ quite significantly – however, we found the variability in mean CDRH3 length within subjects of the same repertoire type insignificant in comparison to the variability between subjects of different repertoire types. We have shown this in a plot and included it as Figure 4- figure supplement 1, and referred to it in the main text

Main

Line 868

The intra-species variance in CDRH3 length is small in comparison to the inter-species difference defined above (Figure 4- figure supplement 1).

3. Section starting at line 154: Is C57BL/6 the most appropriate strain for comparison? Are the constant gene regions from 129S7? Some rationale/discussion should be provided here regarding the genetic background of the Kymouse.

We agree that this is an important point worth rationalising and we have added our reasoning to the appropriate Methods section. Briefly, given that the Kymouse does have some C57BL/6 background, except for the murine constant region which is indeed 129S7, this is still an appropriate comparison and indeed the best comparison available, as the only publicly available large repertoire sequencing datasets are in C57BL/6 and BALB/c mice. Given that the Kymouse is partly bred from C57BL/6, C57BL/6 is the natural choice.

Methods

Line 353

“For a comparator for CDRH3 sequence and structural sharing, we used a previously published dataset of high-depth bulk V_H_ sequencing from C57BL/6 mice (Greiff et al. 2017). This is a reasonable comparator as the Kymouse is bred from a mixture of C57BL/6 and 129S7 strains. The ideal comparator would be 129S7 as the Kymouse constant region genes are derived from the 129S7 strain, however we could not find equivalent publicly available data for this strain. Furthermore, the C57BL/6 strain is best characterised immunogenetically and at the BCR repertoire level (Jackson et al. 2022; Greiff et al. 2017).”

4. Lines 132-136 and 141-145: More detail is needed here. What protocol was used to conduct RT-PCR and generate VDJ libraries? What type of MiSeq libraries? 2x300 bp? Where UMIs used? I presume that PRESTO was also used? This should be stated as well as other parameters used/details regarding sequence read processing.

We apologise for the incompleteness of our description of the sequencing protocol. We have added further detail which we hope is sufficient..

Next Generation Sequencing (NGS) Analysis of Paired V_H_ and V_L_ Sequences from Single-Cell Sorted B Cells Derived from Kymice

For 22 Kymice, the spleen and inguinal lymph nodes were processed to single-cell suspensions as described above before fluorescently-activated cell sorting (FACS) to recover CD19^+^ B220^+^ B cells into individual wells of a 96 well plate. RT-PCR was performed to amplify the V_H_ and V_L_ domains, and standard 2x300bp Illumina libraries were generated before sequencing on an Illumina MiSeq sequencer as previously described (Lee et al., 2014). Unique molecular identifiers (UMIs) were used to group derivatives of the same RNA molecules.

pRESTO was used to process the reads and build UMI consensus groups (Vander Heiden et al. 2014). The Change-O pipeline (Gupta et al. 2015) was then used to process the resultant sequences; naïve BCR sequences were characterized as IgM sequences containing zero nucleotide mutations. In total, 3,175 paired V_H_ and V_L_ sequences were processed from the 22 Kymice. In subsequent analyses comparing gene frequencies, we removed data from seven Kymice where fewer than 100 paired sequences were retrieved, resulting in 15 Kymice contributing to the final dataset. In these 15 Kymice, the average sample size was 202 (range: 105 to 363).

NGS Sequence Analysis of V_H_ Sequences from Bulk Sorted B Cells Derived from Kymice

Bone marrows from the femur and tibia of each Kymouse were processed into single-cell suspension as described above. From these 22 bone marrow samples, seven were FACS sorted to recover CD19^+^ B220^+^ B cells into a single tube. The cells were lysed and RT-PCR was performed to amplify the V_H_ domain, followed by standard 2x300bp Illumina library generation, before sequencing on an Illumina MiSeq. The pRESTO and Change-O pipelines were used to process the sequences generated by the MiSeq sequencers (Vander Heiden et al. 2014; Gupta et al. 2015). IgM sequences with zero mutations were selected for further analysis resulting in a total of 412,493 V_H_ sequences across the seven Kymice (average: 58,928, range: 31,905 –100,240).

5. Lines 144: A description of processing steps leading to the 412,493 sequences would be useful here. (Same comment for human data, line 151).

We have added additional clarification in the relevant sections as to our processing methods.

NGS Sequence Analysis of V_H_ Sequences from Bulk Sorted B Cells Derived from Kymice

Bone marrows from the femur and tibia of each Kymouse were processed into single-cell suspension as described above. From these 22 bone marrow samples, seven were FACS sorted to recover CD19^+^ B220^+^ B cells into a single tube. The cells were lysed and RT-PCR was performed to amplify the V_H_ domain, followed by standard 2x300bp Illumina library generation, before sequencing on an Illumina MiSeq. The pRESTO and Change-O pipelines were used to process the sequences generated by the MiSeq sequencers (Vander Heiden et al. 2014; Gupta et al. 2015). Change-O’s MaskPrimers was used to assign isotypes; germline genes were assigned and identity to germline was calculated using Change-O’s AssignGenes with the Kymouse germline database. All IgM sequences with zero mutations were selected for further analysis resulting in a total of 412,493 V_H_ sequences across the seven Kymice (average: 58,928; range: 31,905 –100,240).

NGS Sequence Analysis of V_H_ Sequences Derived from Human Samples

Buffy coat samples were obtained from ten healthy individuals as described previously (Ghraichy et al., 2021). In the previous study, B-cells were FACS sorted into naïve, marginal zone, plasma and switched memory cell populations. In the present study, we re-analysed a subset of these sequences, which were IgM sequences from the naïve subset of B-cells with zero mutations. Isotypes were used as assigned in the previous study. Germline genes were assigned and germline identity were calculated using Change-O’s AssignGenes with the human reference IMGT database (Gupta et al. 2015; Giudicelli et al. 1997). There was a total of 338,677 sequences (mean: 33,867 per human, range: 20,653 – 48,293).

6. Lines 140 and 149: Should be "FACS"?

We thank the reviewer for noticing this, and we have amended the manuscript accordingly.

7. Section starting at Line 147: Can you be more explicit? Did you generate additional data from the same samples used by Ghraichy et al? Or did you reanalyze data from this study?

We apologise for the lack of clarity here. Indeed, we re-analysed data from the Ghraichy study. We have clarified this.

Line 348:

“In the present study, we re-analysed a subset of these sequences, which were IgM sequences from the naïve subset of B-cells with no mutations. There was a total of 338,677 sequences (mean: 33,867 per human, range: 20,653 – 48,293).”

7. Section starting at line 154: Is C57BL/6 the most appropriate strain for comparison? Are the constant gene regions from 129S7? Some rationale/discussion should be provided here regarding the genetic background of the Kymouse.

We agree that this is an important point worth rationalising and we have added our reasoning to the appropriate Methods section. Briefly, given that the Kymouse does have some C57BL/6 background, except for the murine constant region which is indeed 129S7, this is still an appropriate comparison and indeed the best comparison available, as the only publicly available large repertoire sequencing datasets are in C57BL/6 and BALB/c mice. Given that the Kymouse is partly bred from C57BL/6, C57BL/6 is the natural choice.

Methods

Line 353

“For a comparator for CDRH3 sequence and structural sharing, we used a previously published dataset of high-depth bulk V_H_ sequencing from C57BL/6 mice (Greiff et al. 2017). This is a reasonable comparator as the Kymouse is bred from a mixture of C57BL/6 and 129S7 strains. The ideal comparator would be 129S7 as the Kymouse constant region genes are derived from the 129S7 strain, however we could not find equivalent publicly available data for this strain. Furthermore, the C57BL/6 strain is best characterised immunogenetically and at the BCR repertoire level (Jackson et al. 2022; Greiff et al. 2017).”

8. Line 164: "IGHV" and "IGHJ" are not defined.

We thank the reviewer for noticing this and we believe our changes have made this significantly clearer.

Line 97

“The heavy chain locus consists of variable (V), diversity (D) and joining (J) gene segments, which recombine to form the variable domain of the heavy chain (V_H_). These genes are referred to as the IGHV, IGHD and IGHJ genes respectively. The first two CDRs of the heavy chain, CDRH1 and CDRH2, are encoded by the IGHV gene alone, while the third and most variable CDR, CDRH3, spans the IGHV, IGHD and IGHJ gene junctions.”

Line 106

“Each of the light chain loci, kappa and λ, consist of V and J gene segments but no D gene segments, and both the germline as well as the *recombined light chain variable region* (V_L_) are less diverse than their heavy chain counterparts (Collins and Watson 2018). *These genes are referred to as the IGKV and IGKJ or IGLV and IGLJ genes for the kappa and λ chains respectively: we use IGKLV or IGKLJ to refer to the V or J genes of either light chain locus collectively.”*

9. Line 199: Sometimes "VH" is used, sometimes IGHV, etc. Maybe either choose one convention or be clear about why one or the other is used.

We apologise for the inconsistency, particularly between VH and V_H_. To rectify this, we have changed all instances of VH to V_H_ (and VL to V_L_). Furthermore, we have clarified in the text what we mean by the V_H_ (the variable domain of the heavy chain) and IGHV (the variable gene encoding part of heavy chain variable domain).

Line 97

“The heavy chain locus consists of variable (V), diversity (D) and joining (J) gene segments, which recombine to form the variable domain of the heavy chain (V_H_). These genes are referred to as the IGHV, IGHD and IGHJ genes respectively. The first two CDRs of the heavy chain, CDRH1 and CDRH2, are encoded by the IGHV gene alone, while the third and most variable CDR, CDRH3, spans the IGHV, IGHD and IGHJ gene junctions.”

10. Line 204: What was done in cases for which the germline assignment was ambiguous due to variation not captured in FWR1?

We thank the reviewer for this interesting comment and indeed this has been a consideration. We did not find this to be a problem at the gene level – in benchmarks on other publicly available data where we have masked parts of the FWR1 region, we tested the congruence between Igblast-assigned V germline genes in FWR1-masked vs. unmasked sequences and found that this does not often lead to incorrect germline assignment at the level of genes. Furthermore, given that these sequences are otherwise non-mutated we do not believe that this is a significant limitation on the accuracy of our results.

11. Lines 243-244: It's not clear what data is available on opig? Authors should outline what is provided. It appears to be processed data. Is the raw data available on any other public repository? E.g., EMBL, NCBI?

We agree that our statement about data availability is not clear. As of original submission, the bulk VH data was available on the Observed Antibody Space; we had not made the paired data available, and as OAS only currently accepts 10x data, we have made it available on Zenodo (10.5281/zenodo.7474233). We are unable to make the raw data available.

12. Line 262: You reference Figure 1C before 1B in the text. The panel order should probably be switched. In addition, it looks like the figures in panels 1C and 1D are currently in reverse order. The labels in panel D seem to match panel A, and C appears to match panel B.

We apologise for this mistake and thank the reviewer for noticing this. We have changed the figure so that the panels are in the correct order.

13. Lines 318-319: I believe the authors mean "dendrogram" rather than "phylogenetic tree".

We thank the Reviewer for pointing this out. “Phylogenetic tree” has been substituted with “dendrogram”.

14. Lines 319-320: It is not necessarily the case that the ky mouse and human are the "same" in the number of genes used, which is quite clear from the heatmap in panel 2A. The authors should be more precise in their language. As noted earlier, the Kymouse does not carry a full set of all human IGHV genes but instead carries IGHV genes found on just the human haplotype that was used to construct the mouse. Your data clearly show this, e.g., based on your "outlier" sample, which is probably the only sample in your cohort that carries IGHV4-38-2. Based on the analysis by Lee et al. 2014, it would appear that the Kymouse does not carry IGHV4-38-2. The authors should not overgeneralize.

The reviewer is correct that this was an overgeneralisation and was not accurate: we apologise for this and thank the reviewer for noting this. Indeed, the Kymouse misses 13 IGHV genes which were found in the human datasets. We plot this in Figure 3—figure supplement 1 and detail the 13 IGHV genes which were observed in at least one human subject and not observed in Kymice.

15. Is there a reason the authors did not also compare gene usage profiles for IGK and IGL datasets to control human samples?

This is well-noted – we did not have a compelling reason not to do this besides the generally lower interest in the light chain as a result of its lower diversity and involvement in antigen-engagement. Our only light chain data is from the paired sequencing dataset. We now include analyses of the gene frequencies in this data to paired human data recently released by Jaffe and colleagues. Given that the heavy chain gene frequencies observed in the paired data appeared to correlate well with those observed in the bulk data, we did not perform bulk VL sequencing. The analyses are as below.

Methods

Next Generation Sequencing (NGS) Analysis of Paired V_H_ and V_L_ Sequences from Single-Cell Sorted B Cells Derived from Kymice

For 22 Kymice, the spleen and inguinal lymph nodes were processed to single-cell suspensions as described above before fluorescently-activated cell sorting (FACS) to recover CD19^+^ B220^+^ B cells into individual wells of a 96 well plate. RT-PCR was performed to amplify the V_H_ and V_L_ domains, and standard 2x300bp Illumina libraries were generated before sequencing on an Illumina MiSeq sequencer as previously described (Lee et al. 2014). Unique molecular identifiers (UMIs) were used to group derivatives of the same RNA molecules.

pRESTO was used to process the reads and build UMI consensus groups (Vander Heiden et al. 2014). The Change-O pipeline (Gupta et al. 2015) was then used to process the resultant sequences; naïve BCR sequences were characterised as IgM sequences containing zero nucleotide mutations. In total, *3,*175 paired V_H_ and V_L_ sequences were processed from the 22 Kymice. In subsequent analyses comparing gene frequencies, we removed data from seven Kymice where fewer than 100 paired sequences were retrieved, resulting in 15 Kymice contributing to the final dataset. In these *15* Kymice, the average sample size was 202 (range: 105 to 363).

NGS Sequence Analysis of Published Paired V_H_ and V_L_ Sequences from Naïve Human B cells

As a human comparator for the single-cell Kymouse data, we used a recently published, high-depth paired sequencing dataset, selecting the subset of IgM sequences with zero mutations from naïve B-cells as made available by Jaffe and colleagues (Jaffe et al. 2022). We removed sequences from cells in which there were multiple heavy or light chains. This resulted in 478,177 unique total sequences across four subjects (average: 119,544 sequences per subject; range: 88,631 – 143,293).

NGS Sequence Analysis of V_H_ Sequences from Bulk Sorted B Cells Derived from Kymice

Bone marrows from the femur and tibia of each Kymouse were processed into single-cell suspension as described above. From these 22 bone marrow samples, seven were FACS sorted to recover CD19^+^ B220^+^ B cells into a single tube. The cells were lysed and RT-PCR was performed to amplify the V_H_ domain, followed by standard 2x300bp Illumina library generation, before sequencing on an Illumina MiSeq. The pRESTO and Change-O pipelines were used to process the sequences generated by the MiSeq sequencers (Vander Heiden et al. 2014; Gupta et al. 2015). IgM sequences with zero mutations were selected for further analysis resulting in a total of 412,493 V_H_ sequences across the seven Kymice (average: 58,928; range: 31,905 –100,240).

Results

The Kymouse naïve antibody sequence repertoire is more human-like than murine-like

Using high-throughput paired sequencing we recovered 3,175 full-length paired IgM V_H_ and V_L_ sequences and a further 451,655 full-length unpaired IgM V_H_ sequences from naïve B cells extracted from the spleens and lymph nodes of 22 Kymice.

In order to evaluate the humanness of the Kymouse naïve B cell sequence repertoire, we performed two comparisons: a comparison of the lower-depth single-cell paired sequencing data with a published, high-depth paired human naïve dataset, and a more extensive comparison of the bulk V_H_ sequences to equivalent datasets of 338,677 V_H_ sequences from human naive B cells, and 268,285 V_H_ sequences from C57BL/6 mice.

[…]

We further examined the distributions of CDRH3 and CDRL3 length (Figure 2F). We found that human CDRH3s were on average longer than Kymouse CDRH3s (16.4aa ± 0.01 vs. 14.5aa ± 0.13); there was no significant difference in CDRL3 length (average 9.8aa ± 0.002 vs 9.7aa ± 0.2). To identify possible causes for a difference in CDRH3 length, we compared IGHD gene alignment length and length of nucleotide insertions at the VD and DJ junctions. While there was a small but significant (5% level) difference in IGHD gene alignment length (average 4.6aa ± 0.06 vs. 4.4aa ± 0.08), the largest difference is in insertion length at both the VD and DJ junctions (average 7.8 nt ± 0.01 vs. 4.1 nt ± 0.1, 7.3nt ± 0.02 vs. 3.7nt ± 0.1). A significantly greater proportion of the Kymouse primary repertoire does not have junctional insertions compared to the human repertoire: 19.1% of the Kymouse sequences had no VD insertions, while 8.2% of the human sequences lacked VD insertions. 14.1% of Kymouse sequences had no DJ insertions while 4.5% of human sequences lacked these insertions. Over 10 times as many of the sequences in the Kymouse repertoires lacked junctional insertions compared to human repertoires (3.8% in Kymice vs. 0.3% in humans).

18. Lines 343-350: It would seem appropriate to also compare CDR3 lengths at an individual level, rather than pooling data across all individuals in each species and taking the average. Again, the assumption is being made that all humans come from the same distribution – pooling the data across all human samples, for example, makes it difficult to assess inter-individual variation. It would be worthwhile at least provide plots and summary data for this in the supplement.16. Line 540: "BCR" already defined previously.

We thank the reviewer for this note and have removed the double definition of BCR.

17. Lines 557-559: It could also be worth noting that these transgenic animals may in fact poorly represent the full extent of responses (in terms of diversity) observed across the human population. This will undoubtedly deserve attention as data from these and other transgenic animal models are interpreted in the future. It would be interesting to understand the authors' viewpoints on what uses or limitations the Kymouse (and other similar transgenic animals) might have.

We agree that this is an important consideration, especially given the mounting evidence for considerable unmeasured allelic and haplotypic diversity revealed by the work of Melissa Smith, Guy Yaari, Corey Watson and Yana Safonova among others – as per another comment, we have added a section highlighting this important limitation of transgenic models which is that they cannot emulate the diversity observed in human populations.

Line 1420

“Kymice contain the immunoglobulin haplotype of a single human subject consistent with reduced diversity relative to outbred human B cell repertoires where the considerable allelic and haplotypic diversity observed in human populations has significant effects on the expressed BCR repertoire (Rodriguez et al. 2022). Given the Kymouse’s use of a single haplotype, and differences in the observed germline gene usages, it is likely that the BCR response to antigen stimulus in the Kymouse will represent only a subset of the possible diversity of the outbred human response and this should be consideration in vaccine response modelling.”